# Mind the Gap: Removing the Discretization Gap in Differentiable Logic Gate Networks

**Shakir Yousefi**
ETH Zürich
Switzerland
syousefi@ethz.ch

**Andreas Plesner**[*]
ETH Zürich
Switzerland
aplesner@ethz.ch

**Till Aczel**[*]
ETH Zürich
Switzerland
taczel@ethz.ch

**Roger Wattenhofer**
ETH Zürich
Switzerland
wattenhofer@ethz.ch

## Abstract

Modern neural networks exhibit state-of-the-art performance on many existing benchmarks, but their high computational requirements and energy usage cause researchers to explore more efficient solutions for real-world deployment. Differentiable logic gate networks (DLGNs) learns a large network of logic gates for efficient image classification. However, learning a network that can solve simple problems like CIFAR-10 or CIFAR-100 can take days to weeks to train. Even then, almost half of the neurons remains unused, causing a *discretization gap*. This discretization gap hinders real-world deployment of DLGNs, as the performance drop between training and inference negatively impacts accuracy. We inject Gumbel noise with a straight-through estimator during training to significantly speed up training, improve neuron utilization, and decrease the discretization gap. We theoretically show that this results from implicit Hessian regularization, which improves the convergence properties of DLGNs. We train networks $4.5\times$ faster in wall-clock time, reduce the discretization gap by 98%, and reduce the number of unused gates by 100%. *Equal contribution.

## 1 Introduction

Deep neural networks have reached human-level performance across a wide array of tasks [1, 2]. However, these advances come at the cost of immense computational demands during training and inference, limiting their deployment in many real-world environments [3–6]. This has sparked growing interest in designing models that retain competitive accuracy while being more efficient [7–11].

At their core, all computations on digital hardware reduce to Boolean operations such as AND, OR, and NOT. This motivates the question: *Can we express and execute machine learning models directly in the native language of hardware - namely, logic gates?*

One approach is logic gate networks (LGNs), which replaces arithmetic computations with compositions of discrete logical operations, thereby enabling efficient inference. While inference with LGNs is efficient, training them poses significant challenges. To address this, differentiable logic gate networks (DL-

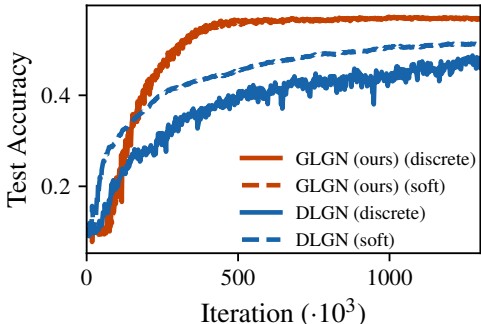

Figure 1: CIFAR-10 test accuracy comparison. Solid and dashed lines show discrete and soft performance, respectively. Gumbel LGNs (GLGN, red) demonstrate faster convergence and minimal discretization gap compared to DLGNs (DLGN, blue).

39th Conference on Neural Information Processing Systems (NeurIPS 2025).

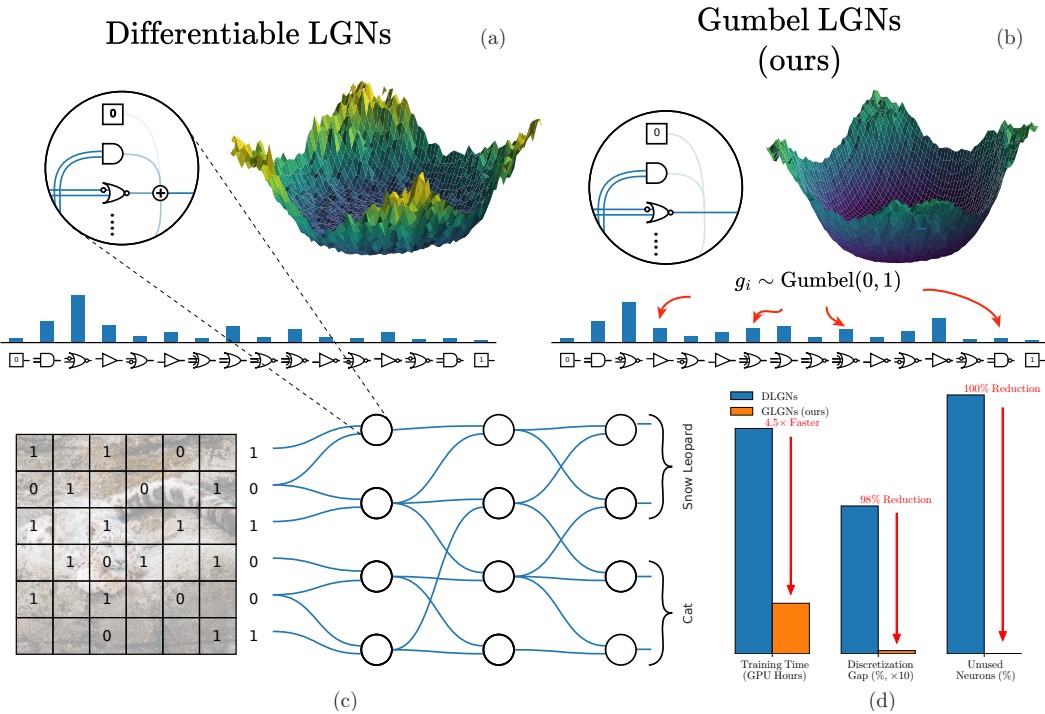

Figure 2: Overview figure. **(a)** DLGNs: Leftmost shows the internal structure of a node. We parameterize each node by weighing the 16 possible logic gates (shown below) and summing their output. This results in a hard, brittle, loss landscape, slowing convergence and increasing the discretization gap. **(b)** Gumbel LGNs: during training, we inject Gumbel noise on the 16 gate weights and select the logic gate with the highest weight. This results in a smoother loss landscape and aligns the training with the network at inference. This results in faster training and reduces the discretization gap. **(c)** Structure of DLGNs and Gumbel LGNs. Each neuron receives two inputs. The nodes in the final layer are aggregated by summation, thus producing class likelihoods. **(d)** Results of using Gumbel LGNs instead of DLGNs. We achieve up to $4.5\times$ faster convergence (in wall-clock time), a 98% reduction in discretization gap, and 100% reduction of unused neurons.

GNs) [7, 9] introduce continuous relaxations of both logical operations and gate selections, allowing the use of gradient-based optimization methods.

Despite their potential, we identify and propose solutions to two major challenges. **(1) Discretization gap**: DLGNs rely on continuous relaxations during training, but final models must be discretized for inference. This mismatch may result in notable degradation, reducing model accuracy by 3% on the same data. **(2) Slow convergence**: While LGNs are efficient at inference time, training DLGNs remains slow due to the reliance on differentiable relaxations, making DLGNs converge substantially slower than standard neural networks.

These challenges are interrelated. The gap arises because the final parameters, after training, must be discretized. Small parameter perturbations can significantly change performance if the loss landscape is sharp [12]. A sharp loss landscape can also cause poor gradient signals, which impact the convergence speed, causing training to take much longer, while a smooth loss landscape can reduce the discretization gap and speed up convergence [13–16]. Our central hypothesis is that *smoother loss landscapes* make DLGN models more robust to discretization and facilitate faster and more stable training. Since the loss landscape is smoother, the gradient signal is better, thus making the networks converge faster. At the same time, the improved gradient signal also causes more neurons to collapse, thus reducing the impact of discretization.

We propose Gumbel Logic Gate Networks (Gumbel LGNs), which inject Gumbel noise into the gate selection process using the Gumbel-Softmax trick. By injecting Gumbel noise into the gate selection process, we introduce stochasticity that flattens the loss landscape, improving the optimization

process and reducing discretization sensitivity. Empirically, we find that Gumbel LGNs exhibit faster convergence and smaller discretization gaps than baseline DLGNs.

To further reduce the discretization gap, we explore a training strategy inspired by techniques from discretization-aware training and neural architecture search (NAS): using continuous relaxations only in the backward pass, while enforcing discrete gates in the forward pass. Although ST estimator may slightly slow convergence, it substantially reduces the discretization gap. Moreover, it aligns training dynamics with inference-time behavior, and as it only influences training, there is no impact on inference.

To our knowledge, no prior work analyzes the discretization gap in DLGNs, and draw formal connections to the smoothness of the loss landscape. NAS and DARTS typically studies explored search spaces up to size $10^{18}$, while our experiments operate over vastly larger parameter spaces, exceeding $10^{3,600,000}$, and show that NAS techniques scale to this setting.

Our contributions are as follows:

- **Empirical validation.** We design and execute experiments showing that Gumbel LGNs both speed up training and boost neuron utilization.

- **Theoretical analysis.** We prove that injecting Gumbel noise into DLGNs smooths their loss landscape by regularizing the Hessian's trace, thereby reducing the discretization gap and accelerating convergence.

- **Practical algorithmic insight.** We demonstrate that employing the straight-through estimator further closes the discretization gap in practice.

## 2   Background

**Logic Gate Networks**   Logic Gate Networks (LGNs) represent an entire network as a composition of discrete logic operations, such as AND, OR, and XOR. LGNs consist of several layers, where each layer contains $n$ *fixed* logic gates, referred to as neurons, that work in parallel. $n$ is the width of the layer. Each neuron in a layer takes as input the output of two (random) neurons in the previous layer. We use the GroupSum operation to get class scores as done by Petersen et al. [7]. For classifying the input into $k$ classes, the neurons in the final layer are grouped into disjoint groups with one group per class. The outputs of the neurons in group $i$ are summed to give the likelihood of class $i$. Mathematically this is expressed as $s_i = \frac{1}{\tau^{\mathrm{GS}}} \sum_{j \in G_i} a_j$, where $G_i$ is the $i$'th group, $a_j$ is the output of neuron $j$, and $\tau^{\mathrm{GS}}$ is a temperature parameter. We then predict the class with the highest score. For more details, see [17].

**Differentiable Training**   Directly searching for the best discrete gate assignments is infeasible due to the size of the search space, so Differentiable Logic Gate Networks (DLGNs) [7, 9] introduce a continuous relaxation. Each of the 16 possible binary gates $h_i(a, b)$ is replaced by a continuous surrogate (e.g. $\mathrm{AND}(a, b) \mapsto a \cdot b$). In addition, each neuron maintains logits $\mathbf{z} \in \mathbb{R}^{16}$, which are initialized using a Gaussian, $\mathbf{z} \sim \mathcal{N}(0, 1)^{16}$. After a softmax, these logits define a probability distribution over gates, and the neuron's output is a weighted sum of the 16 gates:

$$f_{\mathbf{z}}^{\mathrm{soft}}(a, b) = \sum_{i=1}^{16} \frac{\exp z_i}{\sum_j \exp z_j} \cdot h_i(a, b). \tag{1}$$

Ensuring each logic gate maps from a continuous domain $f : [0, 1]^2 \to [0, 1]$. This "soft" network can be trained end-to-end with gradient descent.

**Discretization**   After training, DLGNs are discretized to LGNs by selecting the logic gate with the highest logit value, i.e., it uses $h_i, i = \arg\max_i z_i$. We denote DLGNs evaluated in the differentiable setting (using Equation (1)) as *soft* and otherwise as *discrete*.

**Gumbel-Softmax**   The Gumbel-Softmax trick offers an efficient and effective way to draw samples from a categorical distribution with class probabilities $\pi \in \Delta^k$ [18–21]. Let $g \sim \mathrm{Gumbel}(0, 1)$ distribution if $u \sim U(0, 1)$ and $g = -\log(-\log u)$. We can then draw a sample $z$ from $\pi$ as the

value for index $i$ given by Equation (2).

$$i = \arg \max_j (g_j + \log \pi_j), \quad g_j \sim \text{Gumbel}(0, 1). \tag{2}$$

We can make the argmax operation continuous and differentiable with respect to the class probabilities $\pi_i$, and generate $k$-dimensional sample vectors $y \in \mathbb{R}^k$ using the Softmax with temperature $\tau$ as below:

$$\pi_i^{\text{Gumbel}} = \frac{\exp((\log \pi_i + g_i)/\tau)}{\sum_j \exp((\log \pi_j + g_j)/\tau)}, \quad \pi_i = \sum_{i=1}^{k} \frac{\exp z_i}{\sum_j \exp z_j}, \quad z_i \in \mathbb{R}. \tag{3}$$

## 3 Related Work

**Efficient Neural Architectures**   A significant body of research has focused on designing neural models that maintain high performance while operating within limited computational budgets, e.g., for deployment on edge devices [22–27]. These light models use various methods such as lookup tables [28], binary and quantized neural networks [29–32], and sparse neural networks [33–37].

Of particular interest in this context are differentiable logic gate networks (DLGNs), which have demonstrated state-of-the-art performance in image classification tasks [7, 9], as well as in rule extraction from observed cellular automata dynamics [17]. Our proposed improvements target convergence behavior and are orthogonal to the architectural innovations of the convolutional DLGN variant [9]; hence, we expect them to be transferable without loss of generality. We refrain from comparisons to other efficient neural models, as such benchmarks were already comprehensively addressed in the works mentioned above by Petersen et al. [7, 9].

Kim [38] extend DLGNs by modifying the neuron parameterization introduced in Petersen et al. [7]. Specifically, they normalize the logits and remove the temperature parameter $\tau$. In addition, they employ a straight-through (ST) estimator [39] and injectGumbel noise with a learnable scale parameter. However, Kim [38] do not analyze the discretization gap or study the convergence behavior of DLGNs. Moreover, they draw no formal connection between Gumbel noise and its role as an implicit Hessian regularizer or loss smoothener.

**Differentiable Neural Architecture Search**   Neural Architecture Search (NAS) aims to automate the selection of high-performing model architectures from a large design space [40–42]. While early approaches were computationally expensive, subsequent efforts have focused on improving efficiency [43, 44]. Several works have addressed the issue of train–test performance discrepancies by proposing sampling-based training [45] or regularization techniques that bias architecture selection toward configurations with better generalization [46].

A seminal contribution in this domain is Differentiable Architecture Search (DARTS) by Liu et al. [47], which introduces a softmax-based relaxation over discrete architectural choices, allowing end-to-end optimization through gradient descent. This principle strongly resonates with the soft gate selection mechanism employed in DLGNs.

More recently, Chen and Hsieh [16] reduced the discretization gap of DARTS by introducing Smooth DARTS, which uses weight perturbations through uniform noise or adversarial optimization. These were shown to bias the optimization toward solutions with flatter minima and lower Hessian norm. This technique often referred to as *curvature regularization* reduces sensitivity to sharp local optima and enhances generalization.

**Differentiable LGNs as DARTS**   The works on DLGNs by Petersen et al. [7, 9] do not explicitly draw connections to NAS, but the conceptual similarity is high. Both LGNs and DARTS use softmax-based weighting to choose between multiple candidate functions in a differentiable manner. A key distinction lies in the scale of the search space. Conventional NAS approaches typically explore search space sizes up to $10^{18}$ [48–50],while LGNs operate over exponentially larger spaces—$16^{6 \cdot 64,000} \approx 10^{462,382}$ for MNIST and $\approx 10^{3,699,056}$ for CIFAR-10. This scale is enabled by the simplicity of logic operations, which have no learnable parameters. Thus, DLGNs demonstrate the viability of DARTS at previously unexplored scales.

Conventional NAS frameworks often permit a retraining phase after discretizing the architecture, thereby reducing the discretization gap. LGNs, in contrast, lack such flexibility, as their neurons contain no parameterized operations, and thus the gap persists.

Moreover, in DARTS and related approaches, this training approach favors operations, such as residual connections [51]. Typically, we aim to avoid these residual connections, as they do not increase the models' expressive power [46]. In relation, Petersen et al. [9] finds that their convolutional method with residual initialization mainly converges to residual connections.

**Sharpness-Aware Minimization**    A parallel line of research focuses on improving generalization by minimizing the sharpness of the loss landscape. Motivated by prior theoretical works on generalization and flat minima [14, 52, 53], Foret et al. [15] introduced Sharpness-Aware Minimization (SAM) in Equation (4). This technique explicitly seeks flat minima by optimizing the worst-case loss within a perturbation neighborhood.

$$\min_{\boldsymbol{w}} L_S^{SAM}(\boldsymbol{w}) + \lambda \|\boldsymbol{w}\|_2^2 \quad \text{where} \quad L_S^{SAM}(\boldsymbol{w}) \triangleq \max_{\|\boldsymbol{\epsilon}\|_p \leq \rho} L_S(\boldsymbol{w} + \boldsymbol{\epsilon}), \tag{4}$$

where $L_{\mathcal{S}}(\boldsymbol{w})$ is a loss function over a training set $\mathcal{S}$ of training samples evaluated for model parameters $\boldsymbol{w}$. $p \in [1, \infty[$ is the $p$-norm used (usually $p = 2$) and $\rho > 0$ is a hyperparameter [15]. Since its introduction, SAM has inspired numerous follow-up studies focused on improving computational efficiency [54–59] as well as providing theoretical insights into its efficacy [60–63]. We refer to Appendix D for a detailed description of SAM.

# 4   Gumbel Logic Gate Networks

We introduce *Gumbel Logic Gate Networks* (Gumbel LGNs), which employ discrete sampling of logic gates via the Gumbel-Softmax trick [20, 21] with a straight-through (ST) estimator [39]. While conventional DLGNs maintain a convex combination of gates throughout training and prune to hard selections at inference time, Gumbel LGNs resemble inference-time behavior directly during training by stochastically selecting individual gates per forward pass. We perturb the gate logits with Gumbel noise and select the most probable gate, following the argmax operation (cf. Equation (2)). During backpropagation, the non-differentiable argmax is approximated using the Gumbel-Softmax (Equation (3)), enabling end-to-end training.

This approach is motivated by two observations: (1) Implicit smoothening via noise: injecting Gumbel noise during the forward pass introduces a form of stochastic smoothing, effectively averaging over local perturbations of the loss surface. As we show, this process approximates a curvature-penalizing loss that favors flatter minima and smaller Hessian norm. In addition, this is known to correlate with improved generalization [15, 16]. (2) Inference-time alignment: In DLGNs, the training objective is misaligned with inference behavior, as training relies on weighted combinations of gates that are ultimately discarded. This discrepancy harms generalization. In contrast, Gumbel LGNs train under the same discrete selection mechanism, which is used at inference.

**Training Gumbel LGNs**    As with DLGNs, we model each neuron as a distribution over binary, relaxed logic gates $\mathcal{S} = \{h_1, h_2, \ldots, h_{16}\}$, where each gate $h_i : [0, 1]^2 \rightarrow [0, 1]$ operates on relaxed Boolean inputs. We associate each gate $i$ with logit $z_i$, and the gate has weight $\pi_i^{Gumbel}$ from Equation (3). The output of the neuron with inputs $(a, b)$ is then given below.

$$f_{\mathbf{z}}^{\text{soft}}(a, b) = \sum_{i=1}^{16} \frac{\exp((\log \pi_i + g_i)/\tau)}{\sum_j \exp((\log \pi_j + g_j)/\tau)} \cdot h_i(a, b) = \sum_{i=1}^{16} \pi_i^{\text{Gumbel}} \cdot h_i(a, b), \tag{5}$$

where $\tau > 0$ is a temperature parameter controlling the sharpness of the distribution. As $\tau \rightarrow 0$, the distribution increasingly peaks around the maximum-logit index [20]. This relaxation enables end-to-end differentiability while encouraging the network to commit to discrete logic gates during training.

We employ a straight-through (ST) estimator to bridge the discretization gap between the continuous relaxation used during training and the hard decisions required during inference. In this formulation, each neuron selects a single logic gate in the forward pass via a hard (non-differentiable) choice,

while gradients are estimated through a soft relaxation in the backward pass. See Appendix E for pseudo-code implementation of the training process. Concretely, during the forward pass, we sample Gumbel noise $\mathbf{g} \sim \mathrm{Gumbel}(0,1)^{16}$ and compute:

$$f_{\mathbf{z}}^{\mathrm{discrete}}(a,b) = h_k(a,b) \tag{6}$$

using the gate $h_k$ with maximum perturbed logit. During the backward pass, we use the soft Gumbel-Softmax relaxation (Equation (5)) to compute gradients, effectively treating the hard output as if it were differentiable:

$$\frac{\partial f_{\mathbf{z}}^{\mathrm{discrete}}}{\partial z_i} := \frac{\partial f_{\mathbf{z}}^{\mathrm{soft}}}{\partial z_i}.$$

This ST estimator mechanism encourages the network to make discrete decisions and allows end-to-end optimization via backpropagation. See Figure 2 or Figure 11 in Appendix A for visualizations.

**Implicit Gap Reduction via Gumbel Smoothing.** We present a theoretical result that supports the use of Gumbel perturbations during training. Consider a loss function $\mathcal{L}$, logits $\mathbf{z} \in \mathbb{R}^{16}$, and $\mathbf{g}$ with i.i.d entries $g_i \sim \mathrm{Gumbel}(0,1)$. Adding Gumbel noise with $\tau \in \mathbb{R}$ to the logits can be seen as Monte-Carlo sample of the objective $J(\mathbf{z})$;

$$J(\mathbf{z}) = \mathbb{E}\left[\mathcal{L}(\mathrm{softmax}((\mathbf{z}+\mathbf{g})/\tau)\right].$$

This can be interpreted as a form of stochastic smoothing. This gives us the following lemma:

**Lemma 1** (Gumbel-Smoothing). *Let $\mathcal{L} : \mathbb{R}^{16} \to \mathbb{R}$ be twice continuously differentiable (with Lipschitz Hessian), and let $\mathbf{z} \in \mathbb{R}^{16}, \mathbf{g} \sim \mathrm{Gumbel}(0,1)^{16}$. Consider*

$$J(\mathbf{z}) = \mathbb{E}\left[\mathcal{L}(\mathrm{softmax}((\mathbf{z}+\mathbf{g})/\tau)\right]$$

*and set $\mathbf{a} = \mathbf{z}/\tau$ and $f(\mathbf{a}) = \mathcal{L}(\mathrm{softmax}(\mathbf{a}))$. We then get the expression below.*

$$J(\mathbf{z}) = \mathcal{L}(\mathrm{softmax}(\mathbf{z}/\tau)) + \frac{\pi^2}{12\tau^2}\mathrm{tr}(H_f(\mathbf{z}/\tau)) + O\left(\tau^{-3}\right).$$

*Proof.* See Appendix B. $\square$

Intuitively, by injecting Gumbel noise during training, we encourage the optimizer to find parameters that are robust to small perturbations. This results in flatter loss landscapes and reduces the sensitivity to parameter discretization when switching to inference mode [12]. The expected loss scales with

$$\frac{\pi^2}{12\tau^2}\mathrm{tr}(H_f(\mathbf{z}/\tau)).$$

As the temperature $\tau$ **increases**, the coefficient $1/\tau^2$ **decreases**, reducing the degree of implicit smoothing. We illustrate this with two representative choices:

- **Small** $\tau$ (e.g. 0.1) $\implies 1/\tau^2$ is large $\implies$ large smoothing, flat minima.
- **Large** $\tau$ (e.g. 2.0) $\implies 1/\tau^2$ is small $\implies$ almost no smoothing.

As a result, adjusting the temperature $\tau$ offers a mechanism to control the strength of this curvature-aware regularization, the convergence of the model, and to reduce the discretization gap *implicitly*.

## 5 Empirical Evaluations

Our empirical evaluations focus on CIFAR-10 and CIFAR-100. Due to constrained resources, we limit experiments by default to 48 GPU hours. To ensure a fair comparison, we use the hyperparameters from [7] whenever possible rather than tuning the parameters, such as learning rate, ourselves. Appendix H.1 contains all the default parameters.

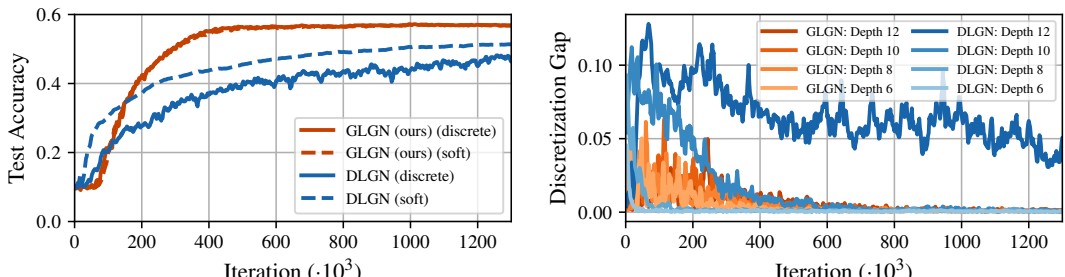

Figure 3: Performance of Gumbel LGNs and DLGNs on CIFAR-10. (**Left**) Test accuracy (width of 256k, depth of 12). (**Right**) Discretization gap for various depths. DLGNs experience larger gaps and slower reduction as the depth increases. In contrast, Gumbel LGNs have consistently low gaps and fast reduction as the network depth increases.

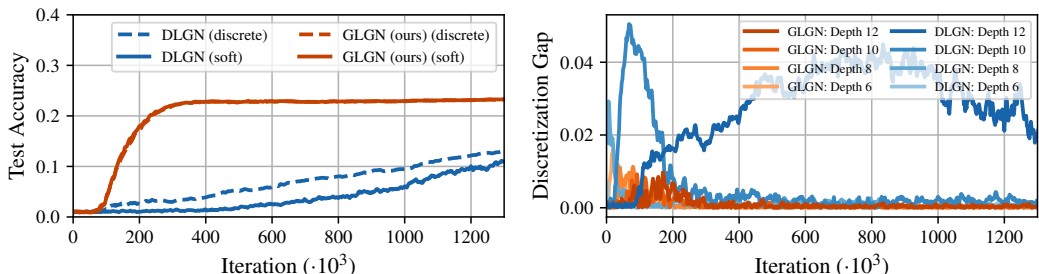

Figure 4: Performance of Gumbel LGNs and DLGNs on CIFAR-100. (**Left**) Test accuracy (width of 256k, depth of 12). (**Right**) Discretization gap for various depths. DLGNs experience larger gaps and slower reduction as the depth increases. In contrast, Gumbel LGNs have consistently low gaps and fast reduction as the network depth increases.

**Discretization Gap**    Figure 3 shows the test accuracy as a function of training iteration for an LGN of depth 12 and width 256k on CIFAR-10; these are the default parameters unless stated otherwise. To quantify the discretization gap, we take the absolute difference between the discretized and soft network accuracy as shown on the right in Figure 3. Gumbel LGN converges much faster than the DLGNs, with virtually no discretization gap. We also measure the runtime and see that Gumbel increases the runtime by roughly 5% per iteration (cf. Appendix K). Combined with runtime results from Table 4 in Appendix K, Gumbel LGN converges[1] $4.75\times$ faster in iterations, making Gumbel LGNs $4.5\times$ faster in wall-clock time to train. Note that the Differentiable LGN still improves after 48 hours.

We perform an additional evaluation on CIFAR-100 using the same settings. On Figure 4, we see that GLGNs already converge at around 400K iterations, whereas DLGNs do not converge. This is consistent with the results, we observe for CIFAR-10.

**Gap Scaling with Depth**    On the right of Figure 3, we see the discretization gap for models of various depths for DLGNs and Gumbel LGNs. As the model depth increases, the expressive power of the networks theoretically increases. The DLGNs experience bigger discretization gaps as the depth increases, while our Gumbel LGNs are stable across depths. Hence, Gumbel LGNs do not struggle to converge as the networks are made deeper. Both methods experience a shift in when the maximum gap occurs. This is expected since increasing the model size usually delays when the accuracy plateaus, i.e., more iterations are needed to converge.

**Shallow Network**    We fix the depth of the network at 6 and increase the width to 2048K. On Figure 5, we see that. Interestingly, the accuracy of the soft DLGN converges within 50K iterations. However, the discretized version does not seem to improve with continued training. We note that the

---

[1]For this, we match Gumbel LGNs' discrete accuracy with DLGNs' maximum discrete accuracy.

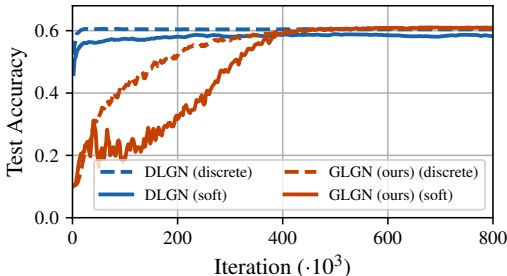

Figure 5: Test accuracy on CIFAR-10 for a shallow, wide network (width 2048k, depth 6).

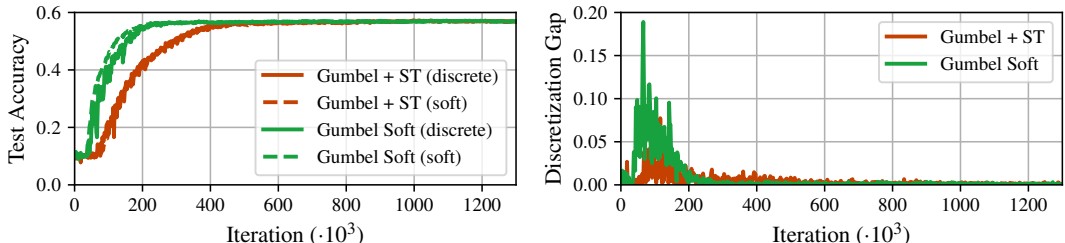

Figure 6: Straight-through (ST) estimator ablation (width of 256k, depth 12). The Gumbel LGNs uses hard gate choices in the forward pass as shown in (2) called the ST estimator. On the left, we show the test accuracy over training iterations; on the right, we show the discretization gap. Gumbel LGNs with ST estimator converge slightly slower in test accuracy, but the discretization gap is smaller.

final accuracy of GLGNs is higher than DLGNs. This suggests increasing the number of neurons through the width also contributes to the discretization gap.

**Straight-Through Estimator Ablation** To better understand the source of gap reduction achieved by the Gumbel-Softmax trick, we perform an ablation to isolate the contribution of the ST estimator. As discussed previously, Gumbel-Softmax combines (i) ST estimation as seen in (2) and (ii) implicit smoothing via Gumbel noise. By disabling the ST path, we aim to identify whether gap reduction primarily stems from the discrete gradient approximation or the added stochasticity. The setup without ST corresponds to DLGNs with noisy logits and is denoted as Soft Gumbel.

In Figure 6, we see that imputing Gumbel noise alone impacts both convergence and discretization compared to DLGNs. However, we observe that including the ST estimator delays convergence for a fixed $\tau$, but further reduces the discretization gap.

**Curvature, Hessian, and Smoothness** As we showed in Section 4, loss minimization for Gumbel LGNs implicitly reduces the curvature of minima by minimizing the trace of the Hessian. The impact of the trace compared to the loss depends on the temperature parameter $\tau$. Thus, we perform an ablation study on the effect of $\tau$, estimate the trace of the Hessian, and visualize the loss landscape.

**Ablation over $\tau$-parameter** We evaluate how varying the temperature $\tau$ affects optimization dynamics. Recall that higher $\tau$ reduces the degree of implicit smoothing, potentially leading to sharper minima and slower convergence. In our experiment, we test $\tau \in [0.01, 2.0]$. We show the results in Table 1 and fig. 7 where we observe a goldilocks zone for the temperature; if $\tau$ is large ($> 1$) or small ($< 0.1$), then the network converges much slower. However, higher temperatures such as 1 seem to converge to slightly better solutions. Although the difference is minor, $< 0.5\%$ when $\tau = 0.25$ goes to $\tau = 1$.

**Hessian Trace Approximation** The Hessian scales quadratically with model size, so direct computations are infeasible for our networks with millions of parameters. Still, we can use iterative methods to approximate the trace, etc. [64–68]. We approximate the trace using Hutchinson's method with

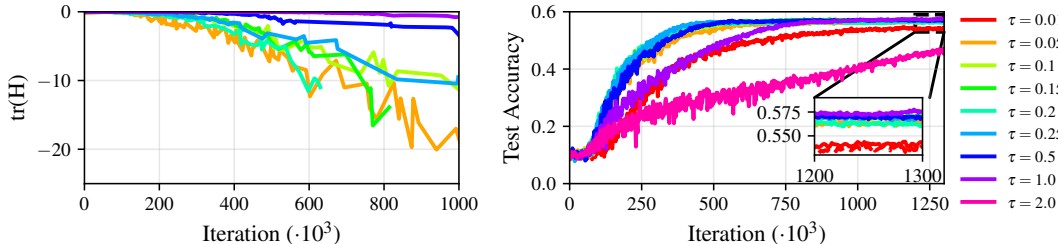

Figure 7: Ablation over the temperature $\tau$ for Gumbel LGNs. **Left)** Estimated Hessian trace using Hutchinson's method. The trace shrinks as $\tau$ decreases, indicating fewer large positive eigenvalues, thus suggesting a flatter loss surface, which may reduce the risk of loss increases when discretizing parameters. **Right)** Test accuracy for the $\tau$-values. We see a goldilocks zone for the temperature, as if $\tau$ is large ($> 1$) or small ($< 0.1$), then the network converges much slower. In the zoomed-in view, we plot the non-discretized view as dashed lines and see that these are similar to the discretized values, i.e., the discretization gap is low for all. Except for $\tau = 0.01$ and $\tau = 2$, there is only a small variation in the final accuracy, as seen in Table 1.

Table 1: Maximum and final test accuracy for the tested $\tau$ values. The iterations column indicates the number of training iterations ($\cdot 10^3$) required to be within $1\%$ of the maximum accuracy. We see that with a medium value of $\tau \approx 0.25$ the network converges much faster than for high values $> 1$.

| $\tau$ | 0.01 | 0.05 | 0.10 | 0.15 | 0.20 | 0.25 | 0.50 | 1.00 | 2.00 |
|---|---|---|---|---|---|---|---|---|---|
| Max accuracy | 0.547 | 0.566 | 0.574 | 0.574 | 0.566 | 0.573 | 0.573 | 0.578 | 0.490 |
| Final accuracy | 0.546 | 0.564 | 0.570 | 0.571 | 0.563 | 0.568 | 0.572 | 0.575 | 0.480 |
| Iterations ($\cdot 10^3$) | 972 | 602 | 632 | 518 | 472 | 440 | 530 | 918 | 1342 |

200 Rademacher random vectors, where each coordinate is independently sampled from $\{-1, +1\}$ with equal probability [67]. Full experimental details are provided in Appendix F.

As shown in Figure 7, decreasing the $\tau$-parameter reduces the estimated Hessian trace. This aligns with the theoretical insights from Section 4; lower $\tau$ values place greater weight on trace reduction. The trace is negative, this is expected: stochastic gradient noise in overparameterized networks tends to systematically lower the expected Hessian trace, biasing solutions toward flatter regions of the loss landscape that may have negative trace values [69–71]. Large negative eigenvalues often vanish, the trace remains influenced by many small eigenvalues, resulting in a negative overall trace.

**Curvature Visualization** To qualitatively assess the loss landscape curvature, we project the high-dimensional parameter space onto two-dimensional subspaces. Following Li et al. [72], we select random directions and interpolate the loss surface along these axes, providing insight into the optimization landscape's geometry around learned solutions. Visualization details are in Appendix G. Figure 8 shows that Gumbel LGN has a visually smoother loss surface.

**Entropy over Logic Gates** Petersen et al. [7] noted that neurons collapse to single gates, but they did not investigate the extent of the collapse. We examine this through neuron entropy by sampling 100k newly initialized neurons and computing the 95% interval. We also estimate expected entropy theoretically (see Appendix J), giving us a baseline distribution for neurons that have not learned. As neurons collapse, their entropy converges to 0. Figure 9 shows that many early Differentiable LGN layers do not collapse, while Gumbel LGN neurons converge with entropies near 0. Defining *unused gates* as those with entropy above the 2.5%-percentile threshold, Gumbel LGNs have 0.00% and DLGNs have 49.81% unused gates, representing a 100.00% reduction by Gumbel.

## 6   Limitations

While Gumbel LGNs demonstrate significant improvements for deeper networks, limitations do remain. Our evaluation focuses primarily on CIFAR-10 and CIFAR-100, with limited exploration of more complex datasets. The temperature parameter $\tau$ requires tuning to balance convergence speed

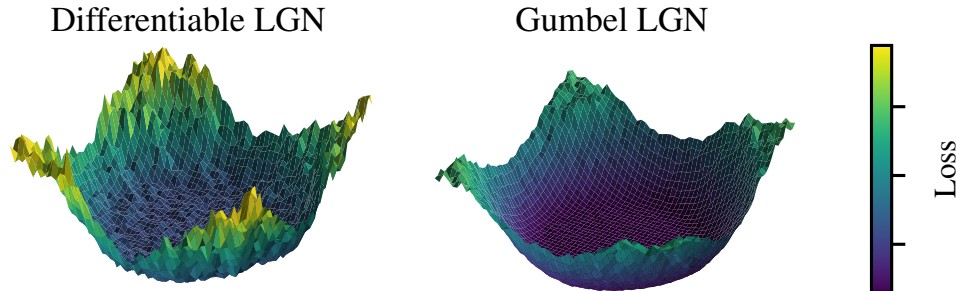

Figure 8: Visualization of loss landscapes. **Left:** Loss landscape of a Differentiable LGN. We see that the landscape is overall noisy. **Right:** Loss landscape of a Gumbel LGN with $\tau = 1.0$. We observe a much smoother loss landscape compared to the DLGNs.

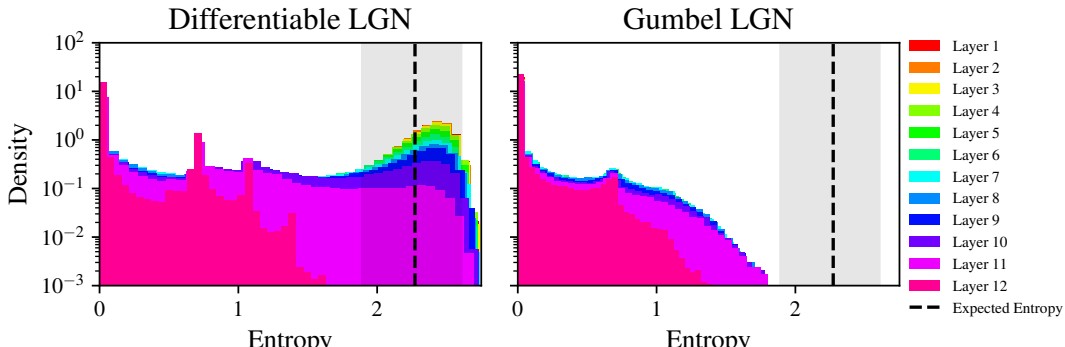

Figure 9: Entropy distribution for neurons in each layer for trained DLGNs and Gumbel LGNs models. The dashed black line indicates the expected entropy (cf. Appendix J) of neurons before training, and the shaded region is the 95% interval computed by sampling. We see that almost all the neurons in Gumbel LGNs have converged, while neurons in early layers of DLGNs still have high entropy.

and accuracy. Finally, our theoretical analysis connects Gumbel noise to Hessian trace minimization under simplifications, but a comprehensive theoretical treatment of the discretization gap remains an open challenge. Moreover, a comprehensive analysis of width scaling and the interplay between width and depth is left for future work.

## 7 Conclusion

We introduced Gumbel Logic Gate Networks (Gumbel LGNs), addressing two critical limitations of DLGNs: slow convergence during training and a large discretization gap. Our theoretical analysis shows that Gumbel noise during gate selection promotes flatter minima by implicitly minimizing the Hessian trace, reducing sensitivity to parameter discretization. Experiments on CIFAR-100 and CIFAR-10 demonstrate that Gumbel LGNs converge up to $4.5\times$ faster in wall-clock time than DLGNs while reducing the discretization gap by $98\%$ and achieving $100.0\%$ improvement in neuron utilization. These advantages become more pronounced with depth, indicating favorable scaling properties. Our improvements are dataset and architecture-independent, and several promising directions remain for future exploration, e.g. adaptive temperature scheduling.

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

# A    Logic Gate Network Visualizations

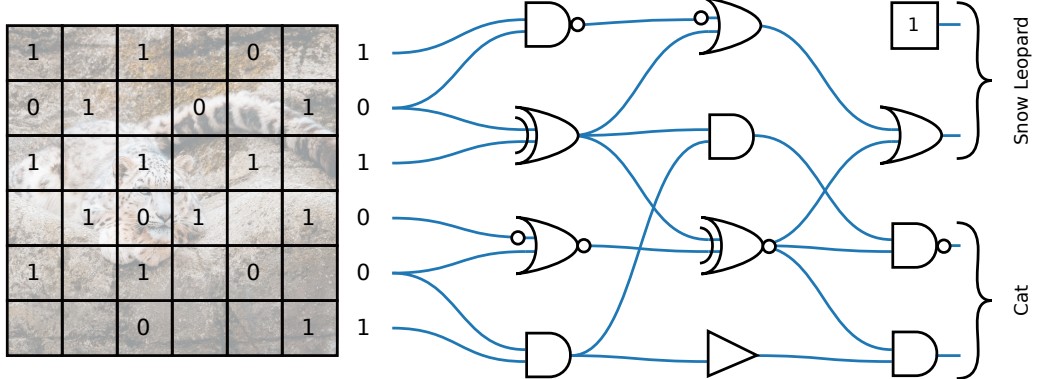

Figure 10: A diagram showing a standard logic gate network (LGN). Each logic gate receives two inputs from the previous layer. The image is first binarized before being passed into the network, and the output neurons are grouped, and each neuron in a group votes whether the image belongs to the group/class.

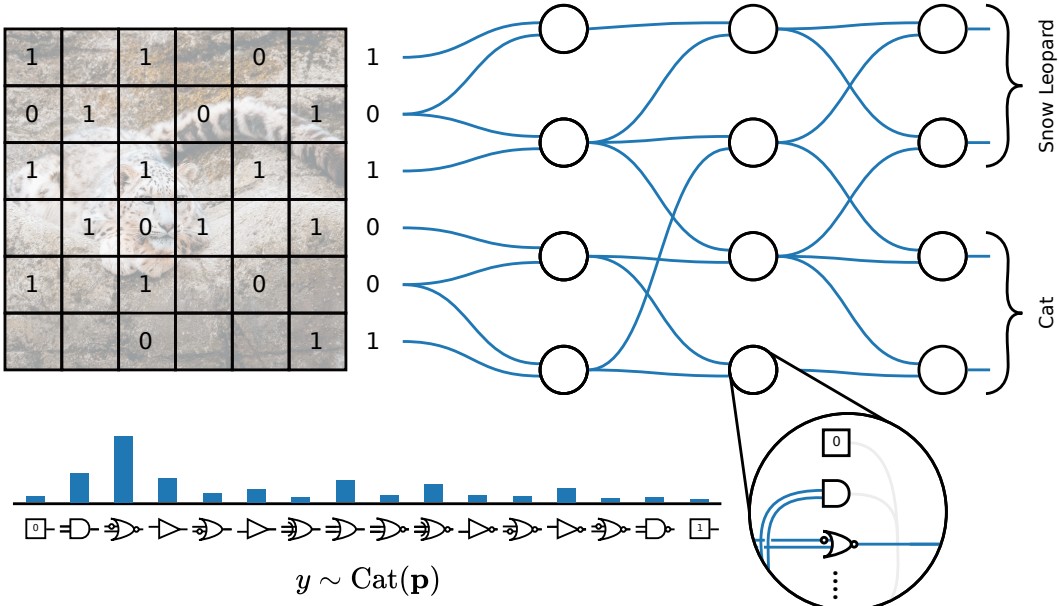

$$y \sim \mathrm{Cat}(\mathbf{p})$$

Figure 11: Forward pass through a Gumbel LGN. The top panel shows neurons producing class scores. **Bottom-left:** categorical distribution $\mathrm{Cat}(\mathbf{p})$ over relaxed logic gates, parameterized by learnable weights $\mathbf{z} \in \mathbb{R}^{16}$. **Bottom-right (zoom-in):** internal view of one neuron. The signal passes through a single selected relaxed logic gate (colors indicate which gate is chosen).

# B    Gumbel Smoothing

The proof of Lemma 1 depends on the translation invariance of the softmax.

**Lemma 2** (Translation-Invariance of Softmax). *Consider logits $\mathbf{z} \in \mathbb{R}^d$, then adding any constant $c \in \mathbb{R}$ to $\mathbf{z}$, $\mathbf{z} + c$, does not change the output of the softmax. Concretely, for any logit $z_i$ we have*

$$\mathrm{softmax}(z_i + c) = \mathrm{softmax}(z_i)$$

*Proof.* Denote $z_i' := z_i + c$, then writing the output of our softmax yields

$$\text{softmax}(z_i') = \frac{e^{z_i'}}{\sum_j e^{z_i'}}$$

$$= \frac{e^{z_i+c}}{\sum_j e^{z_i+c}}$$

$$= \frac{e^c \cdot e^{z_i}}{\sum_j e^c \cdot e^{z_i}}$$

$$= \frac{e^c \cdot e^{z_i}}{e^c \cdot \sum_j \cdot e^{z_i}}$$

$$= \frac{e^{z_i}}{\sum_j e^{z_i}} = \text{softmax}(z_i)$$

$\square$

Restating Lemma 1

**Lemma 3** (Gumbel-Smoothing). *Let $\mathcal{L} : \mathbb{R}^{16} \to \mathbb{R}$ be twice continuously differentiable (with Lipschitz Hessian), and let $\mathbf{z} \in \mathbb{R}^{16}, \mathbf{g} \sim \text{Gumbel}(0,1)^{16}$. Consider $J(\mathbf{z})$*

$$J(\mathbf{z}) = \mathbb{E}\left[\mathcal{L}(\text{softmax}((\mathbf{z}+\mathbf{g})/\tau)\right]$$

*and set $\mathbf{a} = \mathbf{z}/\tau$ and $f(\mathbf{a}) = \mathcal{L}(\text{softmax}(\mathbf{a}))$, then*

$$J(\mathbf{z}) = \mathcal{L}(\text{softmax}(\mathbf{z}/\tau)) + \frac{\pi^2}{12\tau^2}\text{tr}(H_f(a)) + O(\tau^{-3}).$$

*Proof.* Rewriting $J(\mathbf{z})$ in terms of $f$ gives us

$$J(\mathbf{z}) = \mathbb{E}\left[f\left(\mathbf{a} + \frac{\mathbf{g}}{\tau}\right)\right]$$

Consider a second-order Taylor expansion of $f$ around $\mathbf{a}$

$$f\left(\mathbf{a} + \frac{\mathbf{g}}{\tau}\right) = f(\mathbf{a}) + \nabla f(\mathbf{a})^\top\left(\frac{\mathbf{g}}{\tau}\right) + \frac{1}{2}\left(\frac{\mathbf{g}}{\tau}\right)^\top H_f(\mathbf{a})\left(\frac{\mathbf{g}}{\tau}\right) + O(\|\mathbf{g}\|^3/\tau^3)$$

Taking expectations, and recalling that $\mathbb{E}[g_i] = \gamma$, $\text{Var}(g_i) = \pi^2/6$, where $\gamma \approx 0.57721$ is the Euler-Mascheroni constant. we get

$$J(\mathbf{z}) = f(\mathbf{a}) + \left(\frac{\gamma}{\tau}\right)\nabla f(\mathbf{a})^\top \mathbf{1} + \frac{1}{2\tau^2}\left[\gamma^2 \mathbf{1}^\top H_f(\mathbf{a})\mathbf{1} + \frac{\pi^2}{6}\text{tr}(H_f(\mathbf{a}))\right] + O(\tau^{-3})$$

which follows from $\mathbb{E}[g_i^2] = \text{Var}(g_i) + \mathbb{E}[g_i]^2 = \pi^2/6 + \gamma^2$, $\mathbb{E}[g_ig_j] = \gamma^2$ for $i \neq j$.

$$\mathbb{E}[\mathbf{g}\mathbf{g}^\top] = \gamma^2 \mathbf{1}\mathbf{1}^\top + \frac{\pi^2}{6}I$$

and the following trace-lemma

$$\mathbb{E}\left[\mathbf{g}^\top H_f(\mathbf{a})\mathbf{g}\right] = \text{tr}(H_f(\mathbf{a})\mathbb{E}[\mathbf{g}\mathbf{g}^\top]) = \gamma^2 \mathbf{1}^\top H_f(\mathbf{a})\mathbf{1} + \frac{\pi^2}{6}\text{tr}(H_f(\mathbf{a}))$$

Since the softmax is translation-invariant in its input $\mathbf{a}$, we have $\nabla f(\mathbf{a})^\top \mathbf{1} = 0$ and $H_f(\mathbf{a})\mathbf{1} = 0$, so all terms depending on $\gamma$ drop, finally giving us

$$J(\mathbf{z}) = \mathcal{L}(\text{softmax}(\mathbf{z}/\tau)) + \frac{\pi^2}{12\tau^2}\text{tr}(H_f(\mathbf{z}/\tau)) + O(\tau^{-3}).$$

$\square$

Hence, minimizing our stochastic loss implicitly smoothens the curvature by minimizing the trace of the Hessian.

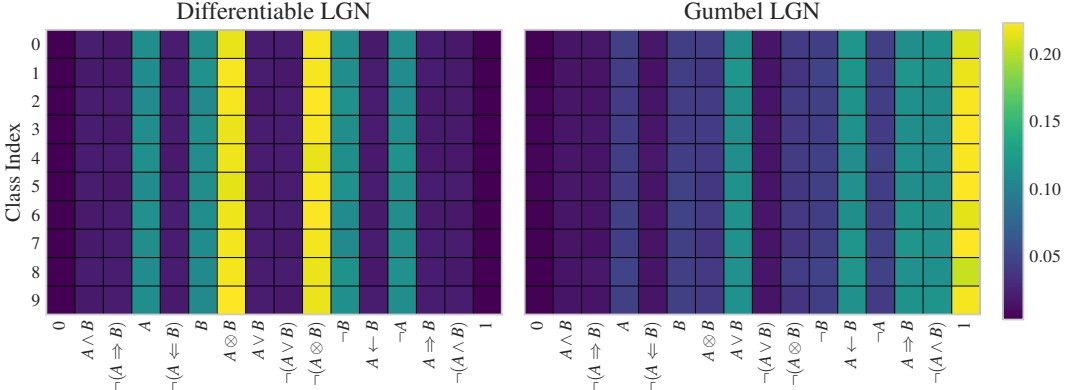

Figure 13: Gate distribution for each class in the CIFAR-10 classifiers. The gates are from the last layer before the groupsum is applied.

# C  Distribution over Logic Gates

**Gate Distribution by Layer**  We look in Figure 12 at the distribution of the logic gates in the final network. Our first observation is that the distributions are far more uniform for Gumbel LGNs in layers 1 to 11 than for DLGNs. At the same time, we see a sharp transition for DLGNs after layer 8. This could match the results in Figure 9, indicating that neurons in DLGNs struggle to converge in all but the final layers.

The "1" gate can be seen as a bias towards specific classes. Notably, Gumbel LGN primarily uses this gate type in the last layer, so we analyze this further.

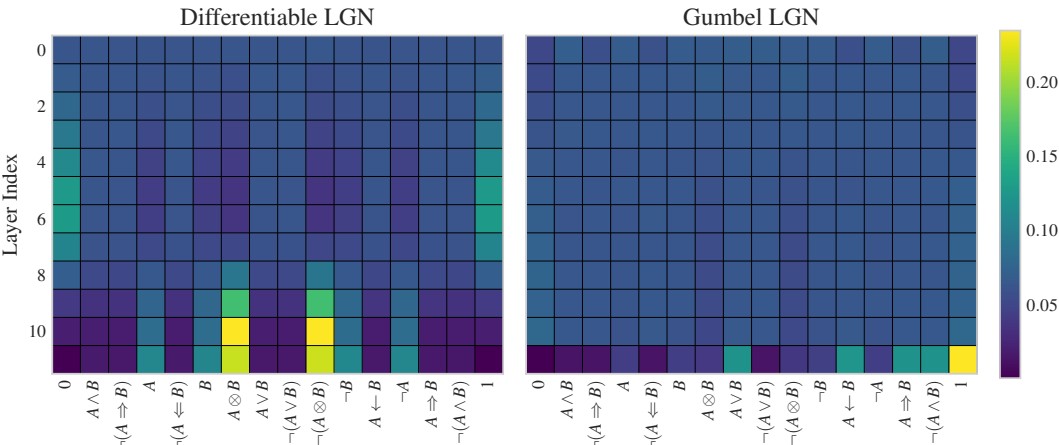

Figure 12: Gate distribution for the gates split by layer for a Differentiable LGN and a Gumbel LGN. Interestingly, the distribution is far more uniform for the Gumbel LGN in layers 1 to 11 than for the Differentiable LGN. In addition, the Gumbel LGN primarily has "1" gates in the final layer, which can be seen as a constant bias towards certain classes. We analyze this further in Appendix C.

**Gate Distribution by Class**  We see in Figure 13 the gate distribution for each of the 10 classes in the last layer. Interestingly, the distributions between Softmax and Gumbel are quite different, but the classes are nearly identical. Since almost all classes have several "1" gates, pruning these would be possible as softmax is translation invariant.

## D  Extended Sharpness-Aware Minimization

A parallel line of research focuses on improving generalization by minimizing the sharpness of the loss landscape. Motivated by prior theoretical works on generalization and flat minima [14, 52, 53], Foret et al. [15] introduced Sharpness-Aware Minimization (SAM). This technique explicitly seeks flat minima by optimizing the worst-case loss within a perturbation neighborhood. Let $L_S(\boldsymbol{w})$ be a loss function over a training set $S$ of samples from a distribution $\mathscr{D}$ evaluated for model parameters $\boldsymbol{w}$.

$$\min_{\boldsymbol{w}} L_S^{SAM}(\boldsymbol{w}) + \lambda \|\boldsymbol{w}\|_2^2 \quad \text{where} \quad L_S^{SAM}(\boldsymbol{w}) \triangleq \max_{\|\boldsymbol{\epsilon}\|_p \leq \rho} L_S(\boldsymbol{w} + \boldsymbol{\epsilon}), \tag{7}$$

where $p \in [1, \infty[$ is the $p$-norm used (usually $p = 2$) and $\rho > 0$ is a hyperparameter [15]. This optimization objective arises from the PAC-Bayesian generalization bound shown in Equation (8) [73], and Foret et al. [15] use it to show Equation (9). Below, $n = |S|$, $k$ is the number of model parameters, $\mathscr{P}$ and $\mathscr{Q}$ are the prior and posterior distributions of the model parameters, respectively. Lastly, the equations hold with probability $1 - \delta$.

$$\mathbb{E}_{\boldsymbol{w} \sim \mathscr{Q}}[L_{\mathscr{D}}(\boldsymbol{w})] \leq \mathbb{E}_{\boldsymbol{w} \sim \mathscr{Q}}[L_S(\boldsymbol{w})] + \sqrt{\frac{KL(\mathscr{Q} \| \mathscr{P}) + \log \frac{n}{\delta}}{2(n-1)}}, \tag{8}$$

$$L_{\mathscr{D}}(\boldsymbol{w}) \leq \max_{\|\boldsymbol{\epsilon}\|_2 \leq \rho} L_S(\boldsymbol{w} + \boldsymbol{\epsilon}) + \sqrt{\frac{k \log\left(1 + \frac{\|\boldsymbol{w}\|_2^2}{\rho^2}\left(1 + \sqrt{\frac{\log(n)}{k}}\right)^2\right) + 4\log\frac{n}{\delta} + \tilde{O}(1)}{n-1}}. \tag{9}$$

Since its introduction, SAM has inspired numerous follow-up studies focused on improving computational efficiency [54–59] as well as providing theoretical insights into its efficacy [60–63].

## E  Training GLGNs

---
**Algorithm 1** Training STE-GLGNs

---
**while** not converged **do**
    Sample Gumbel noise $g \sim \text{Gumbel}(0, 1)$
    Compute soft sample $a' = \text{softmax}((\log \alpha + g)/\tau)$
    Compute hard sample $\hat{a} = \text{one\_hot}(\arg\max a')$
    Forward pass through logic gate network using $\hat{a}$ (with $\text{stop\_grad}(\hat{a} - a') + a'$)
    Compute loss $\mathcal{L}$
    Backpropagate and update $\alpha$ and other parameters
**end while**

---

## F  Estimating Hessian of Loss

Computing the full Hessian of the loss function is not computationally feasible due to the quadratic scaling with the number of parameters. Instead, we use scalable stochastic methods to estimate the trace of the Hessian, which provides useful curvature information. Specifically, we focus on the trace as our Lemma 1 directly minimizes the trace of the Hessian.

**Hessian-Vector Products**  In order to estimate the trace, we will be using Hessian-vector products on the form $Hv$, where $H = \nabla^2 \mathcal{L}(\theta)$ is the Hessian of the loss $\mathcal{L}$ with respect to model parameters $\theta$, and $v \in \mathbb{R}^d$ is an arbitrary vector. While explicitly forming $H$ would require $O(d^2)$ memory and computation, such products can be computed efficiently using reverse-mode automatic differentiation (also known as Pearlmutter's trick) in $O(d)$ time and memory [74].

Given a scalar-valued function $\mathcal{L}(\theta)$, the Hessian-vector product $Hv$ is defined as:

$$Hv = \nabla^2 \mathcal{L}(\theta)\, v = \frac{d}{d\epsilon} \nabla \mathcal{L}(\theta + \epsilon v)\Big|_{\epsilon=0}.$$

This formulation allows for efficient computation using automatic differentiation frameworks, without explicitly constructing the Hessian matrix.

**Trace Estimation via Hutchinson's Method** To estimate the trace of the Hessian, we employ Hutchinson's stochastic trace estimator [67], which approximates the trace of a matrix $H$ as

$$\text{tr}(H) \approx \frac{1}{m} \sum_{i=1}^{m} z_i^\top H z_i,$$

where each $z_i \in \mathbb{R}^d$ is a random vector with zero-mean, unit-variance, i.i.d. entries. The choice of distribution for $z_i$'s affects the variance of our estimator. While both Gaussian and Rademacher distributions satisfy this, Rademacher vectors (each entry sampled from $\{-1, +1\}$ with equal probability) lead to a lower-variance estimator. This is formally shown in [75]. This makes it especially well-suited for estimating curvature efficiently in high-dimensional models.

We use $m = 200$ Rademacher vectors to produce a stable estimate. Each Hessian-vector product $Hz_i$ is computed efficiently using reverse-mode automatic differentiation without explicitly forming the Hessian matrix.

**Choice of Evaluation Points** Since both trace and eigenvalue estimators are noisy and computationally expensive, we evaluate them only at selected points during training. Specifically, we choose points that correspond to monotonically increasing accuracy on the test set.

## G   Loss Surface Visualization

Visualizing the geometry of the loss landscape provides insights into the optimization dynamics during training and the discretization gap. We follow the same procedure as in Li. et al 2018 [76], which constructs a two-dimensional slice of the high-dimensional loss surface by perturbing model weights in random directions.

**Methodology** Let $\theta \in \mathbb{R}^d$ be a parameter vector of a trained model. The goal is to evaluate the loss $\mathcal{L}(\theta')$ over a grid of points

$$\theta'(\alpha, \beta) = \theta + \alpha d_1 + \beta d_2$$

where $d_1$ and $d_1$ are orthogonal directions with $\alpha, \beta \in \mathbb{R}$. We choose $\alpha, \beta$ such that we probe the model in a unit circle, i.e. $(-1, 1)$.

**Direction Sampling** The direction vectors $d_1$ and $d_2$ are generated as follows: Each direction is drawn using a Gaussian distribution $d_i \sim \mathcal{N}(0, 1)^d$. We then normalize $d_i = d_i / \|d_i\|$, such that each perturbation has the same overall scale.

**Orthogonalization** To ensure that $d_1, d_2$ span a meaningful plane, we apply Gram-Schmidt orthogonalization:

$$d_1 \leftarrow d_2 - \frac{\langle d_1, d_2 \rangle}{\langle d_1, d_1 \rangle} d_1$$

This ensures that we span independent, meaningful axes for visualization. This allwos us to visualize the curvature in three-dimensions.

## H   Experimental Configuration

### H.1   Hyperparameters

**Gap Scaling with Depth** we use the same hyperparameters for this experiment for DLGNs and Gumbel LGNs. Specifically, we fix the width of the network to $256K$ neurons and train the network over the depths $\{6, 8, 10, 12\}$. We optimize the network using Adam with a learning rate of $0.01$. Furthermore, the batch size is set to 128 and the final parameter in the GroupSum is set to $1/0.01$. This mirrors the original experimental setup of Petersen et al. [7] for CIFAR-10. Finally, we fix the $\tau = 1$ parameter for the Gumbel noise in the Gumbel LGNs.

| A. Default CIFAR-10 Training | |
|---|---|
| Optimizer | Adam |
| Learning rate | 0.01 |
| Batch size | 128 |
| Depth | 12 (unless varied) |
| Width | 256 k neurons |
| GroupSum scale | 1/0.01 |
| **B. Gap-Scaling (Depth Ablation)** | |
| Depths tested | {6, 8, 10, 12} |
| Width | 256k |
| Other settings | same as (A) |
| **C. Ablation Studies** | |
| Straight-Through vs Soft | depth=12, width=256k, tau=1.0 |
| Temperature sweep | tau in {0.01,0.05,0.1,0.15,0.2,0.25,0.5,1,2} |
| **D. Hessian Estimation** | |
| Trace estimator | Hutchinson, m=200 Rademacher vectors |
| Top-eigenvalue estimator | Power iteration, 200 iterations |
| Evaluation points | checkpoints at monotonic test accuracy |

Table 2: All hyperparameter settings, grouped by experiment.

**Ablation Studies** For both the straight-through and $\tau$-parameter ablations, we use the deepest model from the gap scaling experiment (depth 12, width $256K$) to evaluate each effect in a stress-tested regime. This allows us to isolate the effect of either variant.

## H.2 Implementation

Our code extends the official PyTorch `Difflogic` library by Felix Petersen, i.e., the reference implementation provided alongside the Differentiable LGN paper [7]. During the forward pass we replace the standard `torch.nn.functional.softmax` with a hard Gumbel-Softmax, `torch.nn.functional.gumbel_softmax`, thereby enabling discrete sampling while maintaining end-to-end differentiability.

# I Softmax DiffLogic Performance on MNIST-like Baselines

For all datasets, we use the same model and experiment configs.[2] The models have six layers and a width of 64k. See Table 3 for the results.

Besides the classic MNIST [77, 78], CIFAR-10, and CIFAR-100 datasets [79], we also evaluate EMNIST (balanced and letters) [80], FashionMNIST [81], KMNIST [82], and QMNIST [83]. These are black and white images, as in MNIST, but with other or more classes. We refer the reader to the original papers for details and examples.

---

[2]Thus, the CIFAR numbers are not representative of the optimal performance.

Table 3: The table shows the performance of DLGNs on many datasets. The key takeaway is that the discretization gap for MNIST-like datasets is minimal. The discretization gap is the difference between the discrete and soft performance. The numbers are averaged over five runs.

| | Train | | | Test | | |
| | Accuracy | | | Accuracy | | |
| Dataset | Soft | Discrete | Disc. gap | Soft | Discrete | Disc. gap |
|---|---|---|---|---|---|---|
| CIFAR-10 [79] | 100.0 % | 100.0 % | 0.0 % | 52.04 % | 50.72 % | 1.31 % |
| CIFAR-100 [79] | 83.44 % | 80.39 % | 3.05 % | 23.86 % | 23.1 % | 0.76 % |
| EMNIST balanced [80] | 95.52 % | 94.87 % | 0.65 % | 84.57 % | 84.28 % | 0.29 % |
| EMNIST letters [80] | 98.69 % | 98.3 % | 0.38 % | 91.43 % | 91.04 % | 0.38 % |
| FashionMNIST [81] | 99.02 % | 98.17 % | 0.85 % | 90.37 % | 90.0 % | 0.36 % |
| KMNIST [82] | 100.0 % | 100.0 % | 0.0 % | 97.14 % | 97.0 % | 0.14 % |
| MNIST [77] | 100.0 % | 100.0 % | 0.0 % | 98.33 % | 98.16 % | 0.17 % |
| QMNIST [83] | 100.0 % | 100.0 % | 0.0 % | 98.33 % | 98.17 % | 0.16 % |

## J  Expected Entropy

**Lemma 4.** *The expected entropy of a newly initialized neuron in a Differentiable LGN is $\approx \log 16 - \frac{1}{2} \approx 2.27$.*

*Proof.* A neuron has $n = 16$ gates that it makes a choice over and gate $i$ has (i.i.d.) logit $z_i \sim N(0,1)$ and probability $p_i = \frac{\exp z_i}{C}$ where $C = \sum_{j=1}^{n} \exp z_j$. For the rest of the proof, we assume the number of gates $n$ is not fixed, and show that the expected entropy converges to $\log n - \frac{1}{2}$.

The expected entropy of $p = (p_1, p_2, ...)$ is

$$\mathbb{E}[H(p)] = -\mathbb{E}\left[\sum_{i=1}^{n} \frac{\exp z_i}{C} \log \frac{\exp z_i}{C}\right]$$

$$= \mathbb{E}\left[\log C\right] - \mathbb{E}\left[\sum_{i=1}^{n} \frac{z_i \exp z_i}{C}\right]$$

$$= \mathbb{E}\left[\log \sum_{j=1}^{n} \exp z_j\right] - \mathbb{E}\left[\sum_{i=1}^{n} \frac{z_i \exp z_i}{\sum_{j=1}^{n} \exp z_j}\right].$$

Here, we have as $n \to \infty$:

$$\mathbb{E}\left[\sum_{i=1}^{n} \frac{z_i \exp z_i}{\sum_{j=1}^{n} \exp z_j}\right] = \mathbb{E}\left[\frac{\frac{1}{n}\sum_{i=1}^{n} z_i \exp z_i}{\frac{1}{n}\sum_{j=1}^{n} \exp z_j}\right] = \frac{\mathbb{E}[z_i \exp z_i]}{\mathbb{E}[\exp z_j]}.$$

Using $\mathbb{E}[\exp z_1] = \sqrt{e}$ and $\mathbb{E}[z_1 \exp z_1] = \sqrt{e}$, the above gives us that as $n \to \infty$:

$$\mathbb{E}\left[\log \sum_{j=1}^{n} \exp z_j\right] - \mathbb{E}\left[\sum_{i=1}^{n} \frac{z_i \exp z_i}{\sum_{j=1}^{n} \exp z_j}\right]$$

$$= \log n + \mathbb{E}\left[\log\left(\frac{1}{n}\sum_{j=1}^{n} \exp z_j\right)\right] - \frac{\mathbb{E}[z_i \exp z_i]}{\mathbb{E}[\exp z_j]}$$

$$= \log n + \log \mathbb{E}[\exp z_j] - \frac{\mathbb{E}[z_i \exp z_i]}{\mathbb{E}[\exp z_j]} = \log n + \frac{1}{2} - 1 = \log n - \frac{1}{2}.$$

We only need to plug in $n = 16$ to get the last part. $\qquad\square$

Table 4: Iterations per hour and the relative change from DLGNs to Gumbel LGNs.

|  | Iterations per hour | | |
|  | Softmax | Gumbel | Change |
| --- | --- | --- | --- |
| Depth 6 | 38708 | 38375 | -0.86% |
| Depth 8 | 36292 | 34000 | -6.31% |
| Depth 10 | 32792 | 31167 | -4.96% |
| Depth 12 | 29417 | 27583 | -6.23% |
| Mean |  |  | -4.59% |

## K   Runtime

In Table 4 the number of iterations per hour Softmax and Gumbel LGNs completed while training models for the results in Figure 3. We also calculate the relative difference between the two. Gumbel is slightly slower due to the noise sampling; however, as it converges much faster in terms of iterations, the net effect is that it converges $4.5$ times faster in wall-clock time.

## L   Computational Resources

The experiments were done on an internal cluster with RTX 3090s and RTX 2080 Tis. In total, we have logged 1284 GPU hours for the experiments and testing. A significant part of the compute was spent on exploration.

