# OpenReview forum: "Mind the Gap: Removing the Discretization Gap in Differentiable Logic Gate Networks"
_NeurIPS.cc/2025/Conference — NeurIPS 2025 poster_

### Official Review · Reviewer_FVvS · 2025-06-09

**Clarity:** 3
**Significance:** 3
**Originality:** 2
**Rating:** 3
**Confidence:** 4

**Summary:**

The paper first identifies two important limitations of the current LGNs: slow convergence during training (a simple network can take days to weeks to train) and a large discretization gap between training and inference (i.e., lots of unused neurons). The paper thus proposes to inject Gumbel noise into the gate selection process using the Gumbel-Softmax trick and shows significantly faster convergence time and fewer unused gates.

**Questions:**

The paper studies Logic Gate Networks (LGNs) because they "enable efficient inference". While the paper mainly compares different LGNs on convergence times, is there a comparison on inference times? Because the method can "reduce the number of unused gates by 100%", will it be significantly slower for inference?

**Ethical Concerns:**

["NO or VERY MINOR ethics concerns only"]

**Final Justification:**

This paper focuses on an often overlooked but important direction, LGNs. However, the findings are not that groundbreaking, given that similar ideas have been published by Kim (2023).

**Limitations:**

Although it is stated that the method is "dataset and architecture-independent", the main model is a network with 12 layers and a width of 256k, and there is no implementation for convolutional architectures and on more complex datasets, which is crucial for real-world applications.

**Paper Formatting Concerns:**

No issues.

**Quality:**

3

**Strengths And Weaknesses:**

The paper focuses on Logic Gate Networks (LGNs), an interesting emergent and novel research direction which might be important for the future of DNNs both as a novel neural architecture and as a way for efficient inference. However, the main idea, that using the Gumbel-Softmax for the optimization of LGNs, has already been proposed previously in a recent publication, "Deep Stochastic Logic Gate Networks" (https://ieeexplore.ieee.org/document/10301592), with the same idea, networks, and datasets, decreasing the novelty of the paper.

---

> ### Author Rebuttal · Authors · 2025-07-31
>
> We thank the reviewer for their time and effort in helping to improve the paper. We will address the weaknesses and questions raised in the review.
>
> **Clarifying Experiments and Evaluation Scope**
>
> We would like to clarify that our experiments already evaluate multiple model configurations, not just a single 12-layer, 256k-width network. Specifically, we sweep across four depths (6, 8, 10, 12) (Figure 3) and nine $\tau$-values (0.01–2.0) (Figure 5) at a fixed width of 256k. These sweeps form the basis for our analysis of depth scaling, $\tau$-driven smoothing, and their effect on the discretization gap. We will make this more explicit in the revised draft (Section “Gap Scaling with Depth”) to avoid any ambiguity.
>
> **Clarifying Novelty Relative to Y. Kim, "Deep Stochastic Logic Gate Networks"**
>
> We thank the reviewer for pointing out Y. Kim, "Deep Stochastic Logic Gate Networks," in IEEE Access, vol. 11, pp. 122488-122501, 2023 and will add it to the related works section for completeness. Our paper focuses on analyzing the discretization gap in Differentiable Logic Gate Networks (DLGNs) by framing the problem as Neural Architecture Search and studying how the gap scales, how $\tau$-driven smoothing impacts optimization, and how these dynamics affect pruning and final performance. These aspects are not addressed in Kim (2023), which primarily explores empirical Gumbel-Softmax modifications to improve performance without analyzing gap scaling or optimization dynamics.
> Additionally, Kim (2023) adopts a different neuron parametrization and design that make its regime fundamentally distinct from ours:
> 1) It omits the $\tau$ -parameter (central to Lemma 1 and Fig. 5 in our paper),
> 2) Importantly, normalizes logits. In Kim (2023) this increases entropy in a way that would actually exacerbate the discretization gap, and
> 3) Introduces a learnable Gumbel scaling parameter that alters the Hessian-trace term, complicating any direct comparison to our analysis.
>
> Because of these differences in focus and model regime, our theoretical and empirical analysis addresses a distinct research question.
>
> **On Gumbel Noise and Regularization**
>
> Kim (2023) interprets Gumbel noise as preventing overfitting via Bishop’s result that training with noise approximates Tikhonov/L2-regularization. This derivation assumes zero-mean Gaussian noise, not asymmetric Gumbel noise. Moreover, our results (Fig. 3) show DLGNs continue improving with additional training as the discretization gap narrows, without the accuracy decline typical of overfitting. This indicates that overfitting prevention is **not the primary reason our method works.**
>
>
> **Unused Gates**
>
> Regarding “unused gates”: in both DLGNs and GLGNs, all gates remain in the final LGN architecture and process signals. Many of the neurons in DLGNs maintain high entropy close to initialization or change only slightly by convergence, meaning they are not meaningfully utilized. The final architecture keeps the same gate count, and we expect inference time between the discretized LGNs to match between DLGNs and GLGNs.
>
> **Additional Evaluations**
>
> We agree that broader evaluation is important, and we have now added results on CIFAR-100 and a wider network to support generalization and scalability claims.
>
> On CIFAR‑100 (depths 6–12, width 256k), we observe the same trend as Figure 3: the discretization gap grows with network depth for DLGNs, while GLGNs remain stable. For the largest model, the DLGN has not converged after 1M iterations, reaching a test accuracy/gap of 6.11% / 3.36% points, whereas the GLGN converges in ~400k iterations with 0.23% / 0.0099% points.
> CIFAR‑100’s complexity suggests that larger networks are required, where we expect the discretization gap to become even more significant for DLGNs, making GLGNs critical for scaling.
>
> For a very wide CIFAR‑10 network (depth 6, width 2048k), the DLGN gap remains high (drop of 2.5% points of test accuracy at convergence) and does not shrink significantly over training, whereas GLGNs continue reducing the gap (drop of 0.188% points of test at convergence), consistent with our main results.

---

> > ### Comment · Reviewer_FVvS · 2025-08-05
> > **Thank you for your reply**
> >
> > I appreciate the detailed response, including clarification and new experimental results, from the authors.
> > This paper focuses on an often overlooked but important direction, LGNs. However, the findings are not that groundbreaking, given that similar ideas have been published by Kim (2023). To faithfully highlight the novelty, it would be better to not only mention it in the related works but also benchmark against it to show the difference in performance, if any.

---

> ### Author Response · Authors · 2025-08-07
>
> We thank the reviewer for their response.
>
> >This paper focuses on an often overlooked but important direction, LGNs.
>
> We would like to add that we identify and analyze a key limitation in this subarea, the discretization gap, which, to the best of our knowledge, hasn’t been meaningfully explored in the literature. We believe it is important that this limitation is studied and added to the literature, as it directly impacts the scalability and practical use of DLGNs.
>
> >However, the findings are not that groundbreaking, given that similar ideas have been published by Kim (2023).
>
> We respectfully note that our work addresses a different problem than Kim (2023). We analyze the discretization gap, its scaling, and convergence. None of which are studied in Kim (2023), yet all of which are critical for practical use in this subarea.
>
> >To faithfully highlight the novelty, it would be better to not only mention it in the related works but also benchmark against it to show the difference in performance, if any.
>
> Several important details remain unclear in Kim (2023), e.g. how the learnable parameter $\beta$ is initialized (deterministically or randomly). Kim (2023) reports to follow the same approach as Petersen et. al (2022), yet reports a different learning rate (0.1 versus 0.01). Additionally, the codebase for Kim (2023) is not publicly available, which makes it difficult to faithfully reproduce or benchmark.

---

### Official Review · Reviewer_NbF6 · 2025-06-17

**Clarity:** 3
**Significance:** 3
**Originality:** 2
**Rating:** 4
**Confidence:** 3

**Summary:**

This paper addresses the discretization gap and slow convergence in Differentiable Logic Gate Networks (DLGNs).
The authors adopt Gumbel noise with a straight-through estimator (STE) to train Logic Gate Networks (LGNs) more effectively, aligning training-time behavior with inference-time discretization.
They formally show that injecting Gumbel noise leads to implicit regularization of the Hessian trace, thereby flattening the loss landscape and improving robustness to discretization.
Empirically, the proposed Gumbel LGNs achieve a 4.5$\times$ speedup in training, a 98% reduction in the discretization gap, and 100% neuron utilization on CIFAR-10, with favorable scaling to deeper architectures.

**Questions:**

- While the paper provides a theoretical explanation connecting Gumbel noise injection to Hessian trace regularization, it would be helpful to include a more explicit mathematical formulation quantifying how much flatter the resulting local minima become.

- Logic Gate Networks often admit multiple mathematically equivalent but structurally different networks. In Convolutional Differentiable Logic Gate Networks Section A.3. [4], logic simplification was applied before counting the number of gates. Was a similar logic simplification process applied in this work? If so, how does the number of gates after simplification compare between the default training method and the Gumbel-based method?

[4] Convolutional Differentiable Logic Gate Networks, Petersen et al., NeurIPS 2024.

**Ethical Concerns:**

["NO or VERY MINOR ethics concerns only"]

**Final Justification:**

The paper presents solid theoretical and empirical contributions, particularly in analyzing the effect of Gumbel noise on the loss landscape and demonstrating strong performance improvements. However, the framing of DLGNs as a NAS problem is not novel, and the use of Gumbel-based methods has been well-studied in the NAS literature. While the paper is technically sound, its originality is limited. I therefore maintain a score of 4 (Borderline accept).

**Limitations:**

The authors wrote limitations of this work in the section 6.

**Paper Formatting Concerns:**

None.

**Quality:**

3

**Strengths And Weaknesses:**

### Strengths
- The paper presents strong empirical results, demonstrating significant improvements in training speed, discretization gap reduction, and neuron utilization.
- It provides an explicit theoretical justification connecting Gumbel noise injection to implicit Hessian trace regularization, which clarifies why this technique is helpful.
- The authors conduct a systematic analysis of the discretization gap, including ablations over temperature, Hessian trace estimation, and neuron entropy.

### Weaknesses
- The use of Gumbel noise is a well-established approach in the NAS literature [1,2,3]. Since LGNs can be viewed as an instance of NAS, the application of Gumbel in this context is not novel in itself.

[1] SNAS: stochastic neural architecture search, Xie et al., ICLR 2019.
[2] Searching for A Robust Neural Architecture in Four GPU Hours, Dong et al., CVPR 2019.
[3] DATA: Differentiable ArchiTecture Approximation, Chang et al., NeurIPS 2019.

---

> ### Author Rebuttal · Authors · 2025-07-31
>
> We thank the reviewer for their time and effort in helping to improve the paper. We will address the weaknesses and questions raised in the review.
>
> **Use of Gumbel in NAS.**
>
> The reviewer is correct that these and other papers in the NAS literature have used Gumbel noise before. We will make this clearer in the related work where we already included [1,2,3]. We would also argue that:
>
> 1) To the best of our knowledge, framing DLGNs as NAS is itself a novel and new insight. We hope future LGN work will be inspired by this, and further utilize advancements from NAS literature.
>
> 2) Lemma 1 generalizes beyond DLGNs, and would also theoretically justify why [1,2,3] would have a smoother loss landscape and discretization gap, which are not addressed. This would position [1,2,3] against the broader NAS / Smoothing DARTS literature, e.g. Chen and Hsieh:  Stabilizing Differentiable Architecture Search via Perturbation-based Regularization, 2021 and Zela et. al: Understanding and Robustifying Differentiable Architecture Search, 2020.
>
> 3) The scale of the search spaces are not comparable. Thus, our work demonstrates that the methods have potential to scale to much larger search spaces.
>
>
> **Quantification of the flatness of the minima**
>
> We thank the reviewer for this thoughtful suggestion. We agree that further formalizing how Gumbel noise affects the curvature of local minima is an interesting direction. However, we believe this is challenging theoretically without assuming strong properties (e.g., convexity, L-Lipschitz gradients, or toy quadratic settings) that would not hold in our setting.
>
> Our current analysis (Lemma 1) already shows that Gumbel smoothing reduces the expected trace of the Hessian, and Figures 5–6 empirically reflect this effect. A deeper mathematical treatment of curvature under noise may indeed be more suitable for follow-up work in a broader setting.
>
>
> **On Logic Simplification and Gate Counting**
>
> We did not apply the post‑training logic simplification step described in Appendix A.3 of [4], as the synthesis tool used there is not publicly available. Our results report *raw* gate counts for GLGNs and DLGNs so that differences between the default and Gumbel‑based methods directly reflect neuron
> utilization without additional processing.
>
>
> **Additional Evaluations**
>
> To account for limited evaluation, we have now added results on CIFAR-100 and a wider network to support generalization and scalability claims.
>
> On CIFAR‑100 (depths 6–12, width 256k), we observe the same trend as Figure 3: the discretization gap grows with network depth for DLGNs, while GLGNs remain stable. For the largest model, the DLGN has not converged after 1M iterations, reaching a test accuracy/gap of 6.11% / 3.36% points, whereas the GLGN converges in ~400k iterations with 0.23% / 0.0099% points.
> CIFAR‑100’s complexity suggests that larger networks are required, where we expect the discretization gap to become even more significant for DLGNs, making GLGNs critical for scaling.
>
> For a very wide CIFAR‑10 network (depth 6, width 2048k), the DLGN gap remains high (drop of 2.5% points of test accuracy at convergence) and does not shrink significantly over training, whereas GLGNs continue reducing the gap (drop of 0.188% points of test at convergence), consistent with our main results.

---

> ### Comment · Reviewer_NbF6 · 2025-08-04
>
> Thank you for the thoughtful and constructive rebuttal. I appreciate the clarifications and additional evaluations. I would like to offer a further comment regarding the novelty claim.
>
> ---
>
>
> **(Regarding the novelty of framing DLGN as NAS)**
>
> *To the best of our knowledge, framing DLGNs as NAS is itself a novel and new insight.*
>
> I respectfully disagree with this claim. The DLGN formulation [1,2] itself already builds on the principles of DNAS [3]. Moreover, the logic synthesis community has increasingly adopted NAS methodologies to tackle logic circuit optimization problems [4,5]. In particular, I recommend reviewing [5, Sections 2 and A], which survey machine learning in logic synthesis. Therefore, while the application of Gumbel noise in this context is useful and empirically effective, viewing DLGNs through the lens of NAS is not, in itself, a novel framing.
>
> ---
>
> Besides, I am satisfied with the overall contribution of the paper, and I decided to keep my assessment after reading the author response and other reviews.
>
> ---
>
> References
>
> [1] Deep Differentiable Logic Gate Networks, NeurIPS 2022.
>
> [2] Convolutional Differentiable Logic Gate Networks, NeurIPS 2024.
>
> [3] DARTS: Differentiable Architecture Search, ICLR 2019.
>
> [4] Alan Minko et al. IWLS 2023 Logic Synthesis Contest Submissions, 2023. https://github.com/alanminko/iwls2023-ls-contest
>
> [5] Towards Next-Generation Logic Synthesis: A Scalable Neural Circuit Generation Framework, NeurIPS 2024.

---

### Official Review · Reviewer_iYnB · 2025-07-01

**Clarity:** 2
**Significance:** 2
**Originality:** 2
**Rating:** 2
**Confidence:** 5

**Summary:**

This paper addresses the discretization gap between training and testing of differentiable logic gate networks by adding Gumbel noise to the softmax function during logic gate selection training. The perturbation-based regularization smooths the loss landscape, resulting in more stable gradient signals and faster convergence. By using the Gumbel-Softmax trick, the selection of logic gates can be effectively synchronized between the training and inference phases, thereby reducing the discretization gap.

**Questions:**

1. To isolate the effect of perturbations on the loss function, it is necessary to observe LGNs based on L(softmax(z/τ)) across different τ values. In Lemma 1, τ (tau) is not completely separated, so effect of tau seems mixed in the final loss. In figure 6, can different tau values  show different loss landscapes?
2. As shown in Figure 4, despite a larger discretization gap, faster convergence of test accuracy is observed. Does this result match with the statement in lines 49–50 ("Since the loss landscape is smoother, the gradient signal is better, thus making the networks converge faster.")? Is the choice of backward method (or gradient propagation) more critical for convergence speed than the discretization gap itself? Also, it needs to clarify what Gumbel Soft (discrete) and Gumbel Soft (soft) are.
3. Connecting to the above point, is the reduction of the discretization gap from the use of Straight Through Estimation (STE) or Gumbel Noise (which causes flat curvature)? In the Abstract: "We inject Gumbel noise with a straight-through estimator during training to significantly speed up training,… and decrease the discretization gap." This is ambiguous.
4. Although smaller τ (temperature) leads to faster convergence, the best maximum and final test accuracy is achieved when τ=1 (Table 1). Do these rapid training convergences sacrifice test accuracy due to the perturbations, or could there be another interpretation? With  regularization, does worse generalization performance makes sense? Is it only contributing to faster convergence toward a local optimum?

**Ethical Concerns:**

["NO or VERY MINOR ethics concerns only"]

**Final Justification:**

This paper analyzes the effect of Gumbel-softmax for categorical sampling in logic gate networks, which has already been explored in existing work (Kim, 2023). The difference is that it shows discretization gap analysis and convergence speed along temperature values $\tau$ in the Gumbel-softmax. As the loss function $\mathcal{L}(\text{softmax}(z/\tau))$ in the objective function in Lemma 1 contains $\tau$, the effect of $\tau$ is not completely isolated. Therefore, the effect of $\tau$ appears mixed in the shape of the final loss and convergence speed, even though the authors mainly explain it through Gumbel noise. The authors have failed to explain this clearly. Consequently, their experimental results are misleading and their conclusions are incomplete.

Beyond the Gumbel-softmax setup and STE framework and their motivation in existing work, Hessian-based regularization using the Gumbel-Softmax function has already been explored in "Stabilizing Differentiable Architecture Search via Perturbation-Based Regularization", which further challenges the novelty of this paper's approach.

Thus, I maintain my initial assessment.

**Limitations:**

As noted by the authors, the models should be tested over more datasets.

**Quality:**

2

**Strengths And Weaknesses:**

- **Strengths**:
    - The authors provide a theoretical analysis showing that Gumbel perturbations during training act as a form of Hessian trace regularization, derived using a second-order Taylor expansion of the Gumbel-Softmax function, somewhat inspired by [3,4]. Furthermore, the paper presents a comparison of the loss landscapes between LGN and GLGN.
- **Weaknesses:**
    1. Gumbel-Softmax-based differentiable logic gate networks have been explored in *Deep Stochastic LGN*, where the Gumbel-Softmax formulation is further modified.
    2. In this paper, only a single model is tested (depth: 12, width: 256k), whereas the LGN paper evaluates multiple models with varying depths and widths. It would be more convincing to include comparisons with the model configurations used in Petersen et al.'s LGN paper.
    3. The main contribution of this paper is Lemma 1 (Gumbel-Smoothing), which derives a trace of the Hessian term in the expectation of the Gumbel softmax function. The smoothness of the softmax-based class probability function (Eq. 4) depends on the trace of the Hessian term with a tau (temperature) related weight (in front of the Hessian trace) and also a temperature (tau) inserted into the softmax function. However, the temperature term in the softmax function is neither specified nor explored (in Figure 3).


*Deep Stochastic LGN*:
Y. Kim, "Deep Stochastic Logic Gate Networks," in IEEE Access, vol. 11, pp. 122488-122501, 2023,

---

> ### Author Rebuttal · Authors · 2025-07-31
>
> We thank the reviewer for their time and effort in helping to improve the paper. We address the reviewer’s concerns below:
>
>
> **Clarifying Model Evaluations**
>
> We would like to clarify that our experiments already evaluate multiple configurations, not just a single model. Specifically, we sweep across four depths (6, 8, 10, 12) (Figure 3) and nine $\tau$-values (0.01–2.0) (Figure 5) at a fixed width of 256k. These experiments form the basis for our analysis of depth scaling and $\tau$’s role in loss smoothing and the discretization gap.
> We will make this more explicit in the revised draft (Section “Gap Scaling with Depth”) to avoid any ambiguity.
>
>
> **Clarifying Terminology (DLGN vs. LGN)**
>
> Since our paper centers on the discretization gap, the distinction between Differentiable Logic Gate Networks (Differentiable LGNs, DLGNs), used during training, and fully discretized Logic Gate Networks (LGNs), used during inference, is critical. Some comments (e.g., regarding Lemma 1) appear to reference LGNs when DLGNs were intended. We will make this distinction clearer in the paper and correct mistakes.
>
> **Clarifying Novelty and Relation to Deep Stochastic LGN: Y. Kim, "Deep Stochastic Logic Gate Networks"**
>
> We thank the reviewer for pointing out Y. Kim, "Deep Stochastic Logic Gate Networks," in IEEE Access, vol. 11, pp. 122488-122501, 2023. We will add it to the related papers for completeness and highlight the differences to our paper. While both papers use variations of Gumbel-Softmax, our contributions differ substantially. Our work focuses on analyzing the discretization gap and its scaling behavior, framing DLGNs as a Neural Architecture Search problem. In particular, Kim (2023) does not study the gap or its scaling, nor does it analyze convergence speed or neuron utilization, all central to our theoretical and empirical results.
> Beyond scope, Kim (2023) adopts a different Gumbel-Softmax formulation and neuron parameterization that make its results incompatible with our analysis:
> 1) it omits the $\tau$-parameter (central to Lemma 1 and Fig. 5),
> 2) normalizes logits (as shown in Kim (2023) this increases entropy over the induced distribution, which would worsen the discretization gap), and
> 3) introduces a learnable Gumbel scaling parameter that alters the Hessian-trace term. These choices place Kim (2023) in a different training regime, so our theoretical and empirical results do not apply to their setup.
>
> **On Gumbel Noise and Regularization**
>
> Kim (2023) suggests Gumbel noise prevents overfitting, drawing on Bishop’s result that training with noise approximates Tikhonov / L2-regularization. However, that derivation assumes zero-mean Gaussian noise, not asymmetric Gumbel noise. Moreover, our results (Fig. 3) show DLGNs continue improving with training as the discretization gap narrows, without the accuracy decline characteristic of overfitting. **This suggests that overfitting prevention is not the reason our model works.**
>
> We hope this clarifies the distinctions between our work and Kim (2023). Below, we address the reviewer’s remaining questions.
>
> 1) We believe the reviewer is referring to Differentiable Logic Gate Networks (DLGNs), as clarified in our terminology section, since LGNs denote the discretized (post-training) networks.
> In Figure 6, we visualize the loss landscapes of DLGNs and GLGNs, trained on the setup from Figure 3 (left). We will update the caption to clarify that $\tau = 1$ was used for the DLGN. Even at this baseline $\tau$ value, GLGNs already exhibit a visibly smoother landscape, supporting our core claim about Gumbel-induced smoothing.
> Lemma 1 formalizes this effect, showing that the expected trace of the Hessian scales like $1/\tau^2$, meaning smaller $\tau$ implies stronger implicit smoothing. This is also reflected empirically in Figure 5.
> We agree that additional $\tau$-specific loss visualizations could serve as a useful sanity check. However, we believe that the combination of the $\tau$-sweep (Figure 5), the visual comparison (Figure 6), and our theoretical derivation (Lemma 1) already provides a comprehensive view of the relationship between $\tau$, smoothing, and optimization performance.
> 2) Figure 4 compares training modes in our STE ablation. As stated in lines 235–236, Gumbel Soft refers to a DLGN with Gumbel noise applied to logits or equivalently a GLGN trained without STE. The labels (soft) and (discrete) correspond to the softmaxed network and its fully discretized LGN version. We will clarify this in the figure caption.
> This directly supports the statements in lines 49–50 and aligns with Lemma. Smoothing from Gumbel noise improves convergence. Notably, adding STE slows convergence, confirming that it helps reduce the discretization gap but is not the main driver of faster training.
>
> 3) As shown in Figure 4, Gumbel noise alone significantly reduces the gap between soft and discrete performance, whereas adding STE to Gumbel noise effectively removes it entirely. We will revise the abstract accordingly to reflect this distinction more clearly.
>
>
> 4) From our experiments it does seem to be the case that the models get caught in a local optimum when tau is small. And while the Gumbel noise does significantly help with convergence, GLGNs still have some of the same issues with vanishing gradients as observed in the original works by Petersen et al.
>
>
> **Additional Evaluations**
>
> We agree that broader evaluation is important, and we have now added results on CIFAR-100 and a wider network to support generalization and scalability claims.
>
> On CIFAR‑100 (depths 6–12, width 256k), we observe the same trend as Figure 3: the discretization gap grows with network depth for DLGNs, while GLGNs remain stable. For the largest model, the DLGN has not converged after 1M iterations, reaching a test accuracy/gap of 6.11% / 3.36% points, whereas the GLGN converges in ~400k iterations with 0.23% / 0.0099% points.
> CIFAR‑100’s complexity suggests that larger networks are required, where we expect the discretization gap to become even more significant for DLGNs, making GLGNs critical for scaling.
>
> For a very wide CIFAR‑10 network (depth 6, width 2048k), the DLGN gap remains high (drop of 2.5% points of test accuracy at convergence) and does not shrink significantly over training, whereas GLGNs continue reducing the gap (drop of 0.188% points of test at convergence), consistent with our main results.

---

> > ### Comment · Reviewer_iYnB · 2025-08-04
> >
> > I appreciate the authors' time in clarifying my comments. Unfortunately, several questions remain unresolved and unanswered, while new concerns have emerged.
> >
> > - While the authors emphasize the differences between Kim's Deep Stochastic LGNs (DSLGNs) and their GSLGNs, I respectfully disagree. I have reviewed the authors' responses and compared them with DSLGNs. While DSLGNs employ a learnable temperature parameter $\tau$, the normalization term for Gumbel noise persists. The key distinction lies in whether $\tau$ is fixed during training or not. As the authors state in their conclusion, "adaptive temperature scheduling" is future work. Fundamentally, both methods are modifications of the Gumbel-Softmax mechanism and share the same underlying principles.
> >
> > - DSLGNs explicitly describe the role of Gumbel-Softmax in reducing the sampling gap between softmax-based training and one-hot-based inference. This facilitates better synchronization between training and inference phases, as emphasized in their contribution paragraph.
> > - Furthermore, DSLGNs also employ Straight-Through Estimators (STEs) during training. As reviewer FVvS also noted, these approaches share substantial methodological overlap.
> > - The authors' response remains insufficient regarding $\tau$ into the loss (cross-entropy for classification), the test accuracy results in Figure 5 are more directly influenced by the temperature parameter in the loss function, the Hessian-based perturbation term, and the final shape of the loss landscape. The temperature-scaled softmax affects the logit distribution's uniformity and, consequently, both training dynamics and classification performance. While using Gumbel-Softmax during training does result in a smaller discretization gap and yields a smoother loss landscape, this observation alone does not substantiate the strong claim in the abstract: "This discretization gap hinders real-world deployment of LGNs, as the performance drop between training and inference negatively impacts accuracy."

---

> ### Author Response · Authors · 2025-08-07
>
> We thank the reviewer for their response. We would like to respectfully clarify key points to ensure an accurate evaluation.
>
> Kim (2023) does not mention or analyze the discretization gap, its scaling properties, or convergence behavior. This is the point of our work.
>
>
> >In this paper, only a single model is tested (depth: 12, width: 256k)
>
> This point was addressed in our rebuttal, but not clarified by the reviewer in the response. We would appreciate if the reviewer still believes this is a concern.
>
> Under limitations, the reviewer notes
>
> >As noted by the authors, the models should be tested over more datasets.
>
> We added results for an additional dataset (CIFAR-100). If this remains a concern, we would be grateful for further clarification.
>
> >while new concerns have emerged.
>
> We are unsure which new concerns the reviewer is referring to here. We would also appreciate clarification here.
>
> >... DSLGNs employ a learnable temperature parameter $\tau$
>
> We respectfully note that Kim (2023) does not use a learnable temperature parameter. The learnable parameter ($\beta$ in the notation of Kim (2023)) scales the Gumbel noise, and is not equivalent to our $\tau$-parameter (and the temperature in the canonical Gumbel-Softmax parametrization). We have elaborated further in our general comment.
>
> >The key distinction lies in whether $\tau$ is fixed during training or not.
>
> We believe there may be a misunderstanding about the role of $\tau$ in our work. The reviewer suggests the key distinction is whether $\tau$ is fixed during training, our point is more fundamental: we treat $\tau$ as a conceptual parameter that governs smoothness and the discretization gap, precisely the phenomena that Kim (2023) does not study. Some confusion may stem from the fact that our $\tau$ parameter is called $\lambda$ in Kim (2023). We will make this clearer in the final version. In addition, we kindly remind the reviewer that we raised two additional meaningful architectural differences.
>
> >Fundamentally, both methods are modifications of the Gumbel-Softmax mechanism and share the same underlying principles.
>
> As stated in the rebuttal, we agree on this point. However, one of the main modifications would increase the entropy of the logic gate distributions. We refer to Figure 2 in Kim (2023). High entropy implies a higher divergence between the network during training and inference. This would increase the discretization gap (our Figure 7).
>
> We would gently reiterate the importance of consistent terminology. For example, the reviewer refers to our work as “GSLGN”, whereas we consistently use GLGN or Gumbel LGN.

---

> ### Comment · Reviewer_iYnB · 2025-08-07
>
> I appreciate the authors' time in clarifying my comments. Unfortunately, not all questions have been resolved and answered. Among them, the key issues are:
>
> * Most importantly, the authors do not respond to the critical issue of isolating $\tau$ between the loss function and Gumbel noise perturbation. The lack of isolation makes the authors' interpretation inaccurate and consequently their claims unfounded. Temperature significantly affects the loss function's shape through $\mathcal{L}(\text{softmax}(z/\tau))$, which substantially impacts logit uniformity (acting as a normalization term) and final loss values. Therefore, the discretization gap shown alongside the loss landscape is not solely determined by Gumbel noise. As previously mentioned, the authors must address the $\mathcal{L}(\text{softmax}(z/\tau))$ component. Convergence behavior is also affected by these factors.
>
> * The distinction between temperature $\tau$ and $\beta = \frac{1}{\tau}$ appears negligible (e.g., 1/0.25=4).
>
> * Even though the authors emphasize the difference in formulations, the reviewer is not questioning whether the formulations are actually distinct. The key point is that both methods apply Gumbel-noise-based softmax functions and aim to synchronize one-hot-based sampling between training and inference phases to reduce the discrepancy, termed "discretization gap" by the authors. Gumbel-softmax was not proposed by the authors and is widely used for this purpose. Since the authors have not modified Gumbel-softmax, comparing formulations is irrelevant.

---

> > ### Author Response · Authors · 2025-08-08
> >
> > We thank the reviewer for their response and would like to clarify the role of the $\tau$ parameter in our work to avoid misunderstanding.
> >
> > Our $\tau$ parameter controls only the neuron selection weights. The output class probability softmax temperature is fixed, as specified in Table 2 (“All hyperparameter settings, grouped by experiment”), where the GroupSum scale is set to 1/0.01. While the GroupSum scale indeed affects the shape of the loss landscape, as you highlight. The central focus of our analysis is to isolate and examine the effect of Gumbel noise in the neuron selection on the discretization gap.
> >
> > We understand that temperature choices can, in principle, affect convergence behavior, but in our setup, this effect is separated from the discretization gap measurement. We will clarify this distinction in the final version to avoid possible conflation between the neuron selection temperature and the GroupSum temperature.

---

> > > ### Comment · Reviewer_iYnB · 2025-08-08
> > >
> > > Thank you for the authors' response.
> > >
> > > I continue to ask about the effect of $\tau$ in $\mathcal{L}(\text{softmax}(z/\tau))$ in Lemma 1, as I have raised consistently from my first review through all previous feedback. I believe the authors use a different notation ($\tau^{GS}$) for GroupSum in the manuscript.

---

> ### Author Response · Authors · 2025-08-08
>
> We thank the reviewer for their continued engagement and for raising this point again. We would like to clarify the role of the $\tau$ parameter in Lemma~1.
>
> As stated in the paper, the expected loss scales as
> $\frac{\pi^2}{12\tau^2} \mathrm{tr}\big(H_f(z/\tau)\big).$
> So as the temperature $\tau$ increases, the coefficient $1/\tau^2$ decreases, reducing the degree of implicit smoothing. We illustrate this with two representative choices: Small $\tau$ (e.g., 0.1) $\implies 1/\tau^2$ is large $\implies$ large smoothing, flat minima. Large $\tau$ (e.g., 2.0) $\implies 1/\tau^2$ is small $\implies$ almost no smoothing.
>
> As a result, adjusting the temperature $\tau$ offers a mechanism to control the strength of this curvature-aware regularization, the convergence of the model, and to reduce the discretization gap implicitly.
>
> Table 1 complements this theoretical insight by showing the practical effect of different $\tau$ values on convergence speed and final accuracy.
>
> We would be grateful if the reviewer could kindly specify any further aspects regarding the role of $\tau$ they would like us to clarify.

---

### Official Review · Reviewer_zLS1 · 2025-07-01

**Clarity:** 3
**Significance:** 4
**Originality:** 2
**Rating:** 5
**Confidence:** 4

**Summary:**

This work applies some ideas from other fields to Differentiable Logic Gate Networks (DLGNs), significantly improving their training speed and performance.
Usually, the training of DLGNs involves keeping a distribution over the possible logical gates and employing the associated weighted sum of the gates' outputs as the output of the node.
The core idea of this work is to inject Gumbel noise into the logit values underlying the distribution over the gates and to select the output of the gate with the highest value as the output of the node (no weighted sum). In the backward pass, the non-differentiable "winner-takes-all" operation is approximated using a softmax function.
The authors provide some theoretical justification for the approach and provide experiments that indicate striking improvements over DLGNs.

**Questions:**

In Lemma 1, if we were to replace the $16$ by a general $d \in \mathbb{N}$, what would the thesis look like?

**Ethical Concerns:**

["NO or VERY MINOR ethics concerns only"]

**Final Justification:**

My understanding is that the decisive point here is the novelty relative to *Kim (2023)*.
I maintain my emphatic recommendation of acceptance because:
1. I'm confident the authors can and will fix the blunder of missing *Kim (2023)* in the related works.
2. The results improve the SoTA massively, especially in one of the most notorious and crucial shortcomings of DLGN: their very long convergence times (and they do that while improving accuracy).
3. Relative to *Kim (2023)*:
   - The authors put the reduction of "discretization gap" at the centre of their work, analysing it extensively and *quantitatively*, while *Kim (2023)* only hints at it *qualitatively*.
   - *Kim (2023)* does not study the convergence times, so the massive improvements detected in this work, if present in *Kim (2023)*, would remain unnoticed, limiting the impact of *Kim (2023)* (in this specific regard, that's a good paper).
   - The authors provide their source code, while *Kim (2023)* does not, further amplifying the reproducibility and impact of the work.

I intentionally didn't mention the differences between the methods since **even if the technique proposed was exactly the same as that from Kim (2023),** the points above make me confident that the paper would still be a significant contribution and should absolutely be published.
Put differently, even if this paper were a mere "We investigated *Kim (2023)* from a different angle and found out that it works extremely well", it would still be a great contribution with a meaningful impact in the community.
This is where I diverge from Reviewer FVvS as they mention that "it would be better to not only mention it in the related works but also benchmark against it to show the difference in performance, if any"; I agree that it would be better, but I don't think the this work has no place without this reframe (despite of this divergence, I do appreciate the reviewer's inputs, which are quite constructive).

For those reasons, I, honestly, do not understand Reviewer iYnB's position, which doesn't appear to acknowledge any significance in this contribution.
The reviewer brings many valuable points that would surely improve the paper, but when it comes to motivating the rejection of the work, I don't think the criteria are reasonable.
In fact, the reviewer's opposition is so strong that it gives the impression that they believe not only that the paper should not be accepted for NeurIPS, but that no version of it should be accepted anywhere, given the intersection with *Kim (2023)*.
I truly believe that the contribution is so significant that, even in this case, a preprint version of the paper would likely garner many citations, as its insights would still establish a new reference point for the community.

**Limitations:**

Yes

**Paper Formatting Concerns:**

Are "inline" figures (like Figure 1) allowed?

**Quality:**

3

**Strengths And Weaknesses:**

This work advances the field of DLGNs significantly.
The technique is simple and clean and the results seem impressive.
In fact, I believe the authors could better advertise the improvement in robustness to depth increase as this is an important aspect of their contribution but seems only to be mentioned around line 223.
Presentation is sufficiently clear, and the paper is well-structured with thoughtful guidance to abstracts when suitable.
The theoretical results, though not particularly deep, are a nice addition and provide some justification for the approach.
The rationale behind the experiments is well communicated and the results are clearly presented.

Moving to the weaknesses, the main one seems to be the limited scope of the experiments.
I would go harder on this if the authors had not stated it so clearly as a limitation (the honesty is appreciated).
I also understand that those experiments are quite expensive to run.
Still, the lack of error bars is a major issue, as it makes it hard to assess the significance of the results, which could kill the paper if the improvements were not so pronounced.

Moreover, the claim that unused gates are reduced by 100% can give the wrong impression.
Personally, I did not expect to find much later in the text (around line 273) that this claim is based on defining "unused gates" as those with entropy above the 2.5%-percentile of the entropy distribution.
This threshold appears somewhat arbitrary; if there is a principled reason for choosing this rule, the authors should clarify and justify it in the main text.
Otherwise, the claim should be reconsidered or reworded to avoid potential misunderstanding.

Finally, the use of a fixed dimensionality (16) in Lemma 1 is unusual.
I do understand that this happens to be the number of gates considered in the implementation of LGNs, but I fail to see a reason to limit generality.
Understanding how the result varies with the number of gates adds information on its own.

Below I list some minor points that could help the authors improve the work:
- Although not bad, the introduction of concepts is not particularly good.
I feel like I only got it without much effort because I already had contact with the subject.
For instance, the quick overview of LGNs provided can be improved significantly without taking much more space, if any.
A good example of how to achieve this is already in the references of the paper: [25].
Simply adding an explicit recommendation to that source for an overview of LGNs would already be a meaningful improvement.
- The mathematical typesetting looks a bit amateurish.
Typos are not uncommon and the use of language, which is quite solid elsewhere, becomes clumsy around mathematical expressions.
For some examples, almost all colons in the text are used incorrectly, the statement of Lemma 1 reads a bit odd, and Algorithm 1 is hard to parse since it lacks a proper description of the inputs and their domains (variables need types!).
- More references should be provided.
Not that Yoshua Bengio needs the credit for the straight-through estimator, but it's good practice nonetheless.
There are many other examples of claims that could (and should) be easily backed by references.
- The third equality in the proof of Lemma 2 shouldn't be there.


Overall, I liked the work and I believe it's a clear accept.

---

> ### Author Rebuttal · Authors · 2025-07-31
>
> We thank the reviewer for their time and effort in providing valuable feedback to improve the paper.
>
> **Claim on the number of unused gates**
>
> We agree that the terminology “unused gates” can be easily misinterpreted, as the neurons/gates are still processing the input during training and post-discretization. We use the 2.5% as a heuristic based on a standard two-sided 95% confidence, i.e. neurons whose entropy remain high and statistically similar to initialization. We will clarify this in the revised draft. Perhaps a term like “dead” or “inactive” neuron would better convey this idea.
>
> **Extension of Lemma 1 to general $d \in \mathbb{N}$**
>
> Lemma 1 holds for arbitrary $d \in \mathbb{N}$. Indeed, it would also apply to the general Gumbel-Softmax setting beyond DLGNs. We agree that stating the lemma in its full generality makes more sense - we will change this in the draft.
>
> **Introduction to LGNs**
>
> We thank the reviewer for pointing this out. We agree that the current introduction could be made more accessible, especially for readers unfamiliar with LGNs. In response, we will revise the introductory paragraph to clarify the LGN setup and improve readability without increasing length. Additionally, we will explicitly reference [25] as a recommended resource for readers seeking a more in-depth overview of LGNs. We appreciate the suggestion, as it meaningfully improves the paper’s approachability.
>
> **Mathematical typesetting**
>
> We will carefully revise the mathematical typesetting, addressing the noted issues (colons, Lemma 1 phrasing, and Algorithm 1’s input descriptions). We will ensure all variables are properly typed and all mathematical expressions are clearly presented in the revised draft.
>
> **On References and Citations**
>
> We thank the reviewer for this suggestion. We agree that citing foundational works (e.g., Bengio et al. for the straight‑through estimator, Adam, and other standard methods) will make the paper more polished and complete. We will add these references in the revised draft.
>
>
> **Lemma 2 typo**
>
> We apologise for this mistake and will correct it.
>
> Once again, we would like to thank the reviewer for a thorough and thoughtful review of our paper!

---

> > ### Comment · Reviewer_zLS1 · 2025-08-04
> >
> > I thank the authors for their reply.
> >
> > Starting with a simple note regarding the generalisation of dimensionality in Lemma 1, I was mostly interested in whether the dimensionality would affect the constants in the lemma.
> > However, I was able to disregard the concern by inspecting the proof once more.
> > (The authors should revise that text, by the way. The punctuation and capitalisation are a bit chaotic.)
> >
> > I had a look at *Y. Kim [2023]*, and indeed it seems like a blunder to miss that work.
> > It's very important that the authors include this reference in the related works alongside a thorough discussion about the separation between the two contributions.
> > Without it, it would be fair to reject this paper.
> > Still, seeing the discussion with other reviewers, I'm confident this was an honest mistake and that the authors will address the issue.
> >
> > **Once the fix is made**, I believe the existence of *Kim [2023]* does not critically harm the contribution of this work.
> > In particular, the reference is interested in the related tools for different reasons, and I don't see *Kim [2023]* placing the "Gumbel-Softmax-based" approach as a strong candidate for replacing the original DLGNs in practice.
> > This work does that.
> > It's rare for an approach to improve on previous art so dramatically.
> > I'm confident this work will have a significant impact in the subarea, and I don't see lingering major flaws.
> > Thus, I honestly believe that the technique does perform very well and that it should be accepted for publication.

---

### Official Review · Reviewer_n2V7 · 2025-07-02

**Clarity:** 4
**Significance:** 3
**Originality:** 4
**Rating:** 5
**Confidence:** 3

**Summary:**

This paper introduces Gumbel Logic Gate Networks (Gumbel LGNs) to solve two major issues with existing Differentiable Logic Gate Networks (DLGNs): slow training convergence and a significant "discretization gap" where performance drops after training. By injecting Gumbel noise during the training process, the authors create a smoother optimization landscape. This new method trains 4.5 times faster, reduces the discretization gap by 98%, and eliminates unused logic gates.

**Questions:**

(1) What is the estimated energy efficiency like? \
(2) In addition to the energy efficiency on edge devices, how much does this work advance the practical availability of LGN?

**Ethical Concerns:**

["NO or VERY MINOR ethics concerns only"]

**Limitations:**

yes

**Quality:**

3

**Strengths And Weaknesses:**

**Strengths:**\
(1) The research area of logic gate networks deserves more attention. Analogous to human brain using neurons as basic components, using logic gates as basic components in digital artificial neural networks is a very reasonable research direction. \
(2) The motivation of this work is clear. Hard, brittle loss landscape in existing DLGNs and the inefficiency of neuron usage in inference are significant problems to solve. \
(3) The results demonstrate the effectiveness of Gumble LGN, by being 4.5 times faster than existing DLGNs. \
\
**Weaknesses:**\
(1) The techniques proposed should be more closely bridged to their advantages in practical tasks. This work may have greater impact if some comparisons with standard neural networks can be provided to demonstrate the advantages of Gumble LGN. For instance, with discretization gap and unused neurons minimized, what is the estimated energy efficiency like compared to baselines and traditional NN approaches? Since energy efficiency is pivotal in edge device deployment.

---

> ### Author Rebuttal · Authors · 2025-07-31
>
> We thank the reviewer for their time and effort in helping to improve the paper. We will address the weaknesses and questions raised in the review.
>
> **Advantages of GLGNs over NNs.**
>
> In both DLGNs and GLGNs, all gates remain in the final LGN architecture and are active during inference. Thus, the benefits of logic-gate-based computation (e.g., energy efficiency) apply equally to LGNs induced by either method.
> We refer to estimates from Petersen et al. [2], where a CLGN deployed on a Xilinx XC7Z045 FPGA performs inference on CIFAR‑10 in 9 ns per image achieving 60.38% test accuracy. Assuming a peak power consumption of 15 W, this corresponds to approximately 135 nJ per image. Although the CLGN architecture differs from our models, it shares the same neuron parameterization as DLGNs, so we expect a Gumbel-parameterized CLGN to have comparable inference-time energy characteristics.
>
> **On Advancing the Practical Availability of LGNs**
>
> As the reviewer notes, faster DLGN training directly advances the practical availability of LGNs. Our experiments also show that the discretization gap scales with architecture size and the number of logic gates. For more complex tasks, where larger and more expressive networks are needed, Gumbel LGNs makes scaling LGNs more feasible with less prohibitive training costs.

---

### Author Response · Authors · 2025-08-07
**Novelty compared to Kim (2023)**

We thank the reviewers for their continued engagement. Two reviewers have raised concerns regarding the novelty compared to Kim (2023). In the final version, we will add a detailed discussion clarifying the conceptual and methodological differences between our contribution and theirs. We summarize the differences below in terms of general scope and neuron parametrization. As a final note, we comment on reproducing Kim (2023).


**General Scope**

We respectfully note that our work studies a different problem than Kim (2023). While both use Gumbel-Softmax based mechanisms, our focus is on the discretization gap, a key limitation in DLGNs that impacts scalability and convergence. To the best of our knowledge, the discretization gap, its scaling behavior, and convergence dynamics have not been studied in prior work, making our contribution distinct and practically important.


**Neuron Parametrization**

For clarity, we present both our formulation and Kim (2023)’s using aligned notation. Let $ z_i $ denote the logits, $h_i$ the gates and $g_i$ sampled from $\text{Gumbel}(0,1)$. Our soft relaxation is:

$$ f(a, b) = \sum_{i=1}^{16} \frac{\exp\left( (z_i + g_i)/\tau \right)}{\sum_{j=1}^{16} \exp\left( (z_j + g_j)/\tau \right)} \cdot h_i(a, b) $$

Kim (2023) instead uses:

$$ f(a, b) = \sum_{i=1}^{16} \frac{\exp\left( z_i /  \| \| \mathbf{z} \| \| + \beta \cdot g_i \right)}{\sum_{j=1}^{16} \exp\left( z_j /  \| \| \mathbf{z} \| \| + \beta \cdot g_j \right)} \cdot h_i(a, b) $$

where $\beta$ is a learnable parameter, scaling the *noise*.

We believe the differences are substantive. 1) The $\tau$ parameter is omitted. 2) The logits are normalized. 3) the noise is scaled by a learnable parameter.

Conceptually, 1) is important as it offers insights into the smoothness of the loss and the discretization gap (used in Lemma 1). 2) and 3) increase the entropy of the neuron as seen in Figure 2 of Kim (2023). The distribution in their toy-example has an entropy of approximately 1.29 nats, close to the maximum of 1.386, indicating that multiple logic gates are active during training. This high entropy implies a substantial divergence between training and inference behavior, which increases the discretization gap (as shown in our Figure 7).


**Reproducing results of Kim (2023)**

Several important details remain unclear in Kim (2023), e.g. how the learnable parameter $\beta$ is initialized (deterministically or randomly). Kim (2023) reports to follow the same approach as Petersen et. al (2022), yet reports a different learning rate (0.1 versus 0.01). Additionally, the codebase for Kim (2023) is not publicly available, which makes it difficult to faithfully reproduce or benchmark.

---

### Decision · Program_Chairs · 2025-09-17

**Decision:**

Accept (poster)

**Comment:**

This paper tackles the critical issues of slow convergence and the "discretization gap" in Differentiable Logic Gate Networks (DLGNs), where performance drops when moving from continuous training relaxations to discrete inference-time models. The authors propose injecting Gumbel noise via a straight-through estimator, providing a novel theoretical analysis that links this approach to implicit Hessian regularization and a smoother loss landscape. The resulting empirical gains are substantial, demonstrating a 4.5x training speedup and a 98% reduction in the gap.

The main point of contention during review was the novelty relative to Kim (2023), which also applied a Gumbel-Softmax approach to LGNs. However, based on the discussion, I believe that this paper's contributions are distinct and significant due to its explicit focus on analyzing and solving the discretization gap, its novel theoretical framing, and its impressive empirical results that dramatically improve the practicality of LGNs. Other concerns that were raised were during the discussion were the limited experimental scope. The authors ran additional experiments on CIFAR-100 and on a wider network to address this.

This paper was discussed in detail with the senior AC, and we decided to accept the paper. Note to the authors: please fully integrate the discussion of Kim (2023) into the final manuscript as promised. Furthermore, this final version should also include the additional empirical results provided during the rebuttal and consider revising its claims about the novelty of framing DLGNs as a NAS problem to accurately situate the work within the existing literature.